# Trivariate copula to design coastal structures

Olivier Orcel[1], Philippe Sergent[1], François Ropert[1]

[1]Cerema, Margny-Lès-Compiègne, 60280, France

*Correspondence to*: Philippe Sergent (philippe.sergent@cerema.fr)

**Abstract.** Some coastal structures must be redesigned in the future due to rising sea levels caused by climate change. The design of structures subjected to the actions of waves requires an accurate estimate of the long return period of such parameters as wave height, wave period, storm surge and more specifically their joint exceedance probabilities. The simplified Defra method that is currently used in particular for European coastal structures makes it possible to directly connect the joint exceedance probabilities to the product of the univariate probabilities by means of a single factor. These schematic correlations do not, however, represent all the complexity of the reality because of the use of this single factor. That may lead to damaging errors in coastal structure design. The aim of this paper is therefore to remedy the lack of robustness of these current approaches. To this end, we use copula theory with a copula function that aggregates joint distribution functions to their univariate margins. We select a bivariate copula that is adapted to our application by the likelihood method. In order to integrate extreme events, we also resort to the notion of tail dependence. The optimal copula parameter is estimated through the analysis of the tail dependence coefficient, the likelihood method and the mean error. The most robust copulas for our practical case with applications in Saint-Malo and Le Havre (in Northern France) are the Clayton copula and the survival Gumbel copula. The originality of this paper is the creation of a new and robust trivariate copula with an analysis of the sensitivity to the method of construction and to the choice of the copula. Firstly, we select the best fitting of the bivariate copula with its parameter for the two most correlated univariate margins. Secondly, we build a trivariate function. For this purpose, we aggregate the bivariate function with the remaining univariate margin with its parameter. We show that this trivariate function satisfies the mathematical properties of the copula. We finally represent joint trivariate exceedance probabilities for a return period of 10, 100 and 1000 years. We finally conclude that the choice of the bivariate copula is more important for the accuracy of the trivariate copula than its own construction.

## 1 Introduction

The design of coastal structures requires the multiplicity of variables and their degree of correlation to be taken into account. We must therefore address the lack of robustness in the modelling procedure of the dependencies between the different variables characterizing the sea state (Sergent *et al.*, 2014; Hawkes, 2005) such as wave height $H$, wave period $T$ and storm surge $S$. The design of coastal structures is based in particular on the return periods of wave overtopping or of armour damage (Ciria *et al.*, 2007). Since the applications on wave overtopping and armour damage depend on the parameters of the coastal structure, we will not deal with the return periods of these quantities. The aim of this paper is however to improve the methods of estimating them in order to avoid costly and inappropriate decisions (Li *et al.*, 2008). To this end, we provide accurate estimates of the correlations between the variables $H$, $T$ and $S$ and obtain reliable return period estimates. Currently, in reference manuals such as the Rock Manual (Ciria *et al.*, 2007), it is recommended that a factor be applied to the product of univariate survival functions in order to determine the joint period. This method is named the simplified Defra method.

Copulas are mathematical tools for modelling the dependence structure of several random variables. The theory of copulas was developed by the mathematician Sklar (1959). The copula is a written form of the joint distribution function that provides all the information on the dependency structure. The recent interest in copulas started in financial risk management and insurance. Its use in environmental science especially concerns hydrology with the works for example of De Michele and

Salvadori (2003), Favre *et al.* (2004), Grimaldi and Serinaldi (2006), Genest and Favre (2007), Zhang and Singh (2007),
Aghakouchak *et al.* (2010), Lee *et al.* (2013), Chang *et al.* (2016).

In coastal engineering, in order to estimate the probability of failure of coastal or offshore structures caused in particular by the critical appearance of the combinations of parameters during a storm, Salvadori *et al.* (2007) use a copula in order to link the intensity of storm surge to its duration. Using the copula theory, Hawkes (2005) obtains, for example, the set of pairs of variables wave height *H* and surge *S* for a given return period. The bivariate return period can be generalized to the multivariate
case (Charpentier, 2014).

In this paper we propose the use of copulas to take into account the dependence between three variables *H*, *T* and *S*. We want to show the relative importance of the choice of the copula family and of the copula construction. Tiloy et al. (2020) illustrated the importance of having a range of bivariate models when attempting to capture interrelations between pairs of hazards. In this paper we compare at the same time the choice of the copula family and the choice of the copula construction. Copulas
aggregate easily two random variables. The construction of a trivariate copula requires specific attention as stated by Nelsen (1985). The purpose of this article is the creation of a new trivariate copula and the evaluation of its robustness. In the literature the Chakak and Koehler (1995) method is commonly used and in particular by Joe (1997) and Salvadori *et al.* (2007). This method is based on bivariate conditional distributions and requires the use of three bivariate copulas. The method has a compatibility problem. There is indeed no guarantee that the method gives the same result when the order of variables is
changed. Corbella and Strech (2013) study trivariate copula based on storm magnitude, storm duration and wave height. They show that the fully nested method of creating hierarchical copulas provides the best results for their case study compared to Chakak and Koehler (1995) and conditional mixture. The latter method is similar to Chakak and Koehler (1995). The three dimensional distribution is obtained from the conditional distributions through an integration.

Aas and Berg (2009) propose copula construction with conditional sets : the pair copula construction (PCC). Gouldby et al.
(2014) propose also a methodology for deriving extreme nearshore sea conditions for structural design with waves, winds and sea levels as offshore variables using also conditional distributions. This model is referred to as a conditional extreme model in Tiloy et al. (2020). PCC provides a powerful tool to construct flexible multivariate distributions which can be used to model complex dependencies. This method often performs better than other methods of construction of trivariate copula. Jane et al. (2020) show for example with multivariate statistics between rainfall, ocean-side water level and groundwater level, that PCC
better captures the dependence than any of the five tested standard higher dimensional copulas. Despite its performance, we did not use PCC for two reasons. Our objective is firstly the construction of a trivariate copula that can be easily used by coastal engineers. PCC requires complicated calculation of full conditional probabilities and the construction of vine trees. Secondly, the bivariate copulas that are selected as the most promising in our application are Archimedean copulas. If they satisfy some properties, these Archimedean copulas enable simpler methods of construction of multivariate copulas.

Opting for a balance between accuracy and complexity, we propose to use a fully nested hierarchical trivariate copulas and to test the sensitivity of the results to the method of construction and to the choice of the copula. Showing that Archimedean copulas give the best results for bivariate copulas, we keep them for trivariate copulas. We can then adopt a fully nested hierarchical copula.

The paper is divided into three parts. In a first part, we define the theory by presenting, partly in appendix, the marginal
distribution, the recommended method of the Rock Manual, the copula, the bivariate copula, the tail dependence, the survival copula, the trivariate copula and contours of equal joint exceedance probability for different return periods. We obtain a bivariate copula and the copula parameter by the method of maximum likelihood and the method of the error. We show that the trivariate function that is obtained satisfies the mathematical properties of a copula. In a second part, we present contours of equal joint exceedance probability for applications at the ports of Le Havre and Saint-Malo (Northern France) with bivariate
copulas corresponding to different return periods. We select Clayton copula and survival Gumbel copula as the most robust survival copulas for our coastal engineering based applications. Finally, in a third part, we apply trivariate copulas in Le Havre.

## 2 Theoretical approach

The notations and the main notions of copula for a bivariate distribution function are recalled in appendix A. In order to determine the return period of events that lead to wave overtopping or armour damages, we choose to use survival functions.

As mentioned by Serinaldi (2015), this option is not unique and will lead to a specific return period that is denoted by $T_{AND}$. For two random variables, $T_{AND}$ is directly related to the bivariate survival function $\bar{F}_{XY}(x, y)$ that is also noted $P(X > x, Y > y)$ in appendix A.

We present here the sets of data on the sites, the selection of the best bivariate copula and the construction of trivariate copulas.

### 2.1 Sets of data

The approach is applied in two ports in Northern France, Saint-Malo and Le Havre that are presented in Figure 1.

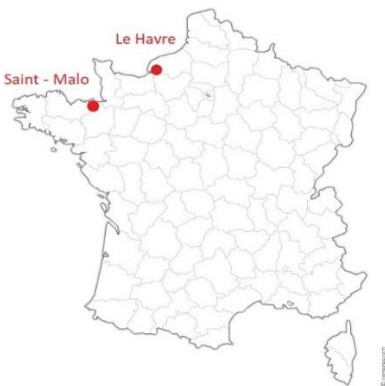

**Figure 1 :** The Saint–Malo and Le Havre sites.

To characterize oceanic forcing, we introduce three random variables, wave height $H$, wave period $T$ and storm surge $S$. The wave height is the significant wave height that is noted $H$ in order to simply the notation. By convention, the random variables are written in capital letters and the realizations of these random variables are written in lowercase ($h$, $t$, $s$). The wave data are derived from the Anemoc numerical database, which reconstructs sea conditions over a period of nearly 25 years by hindcast. Sea levels come from the tide gauges of various ports. The storm surges are deduced from these observations by removing

from the water level the value of the astronomical tide obtained using the Shom Predit software. This software also allows to obtain the density of high tides.

As the study focuses on the integration of tidal range in a macrotidal environment in the calculation of the probability of joint occurrence of waves and water levels, the used data are those of waves and surges taken at high tide. The sample is made of

706 events per year using the same definition as in the Rock Manual. In coastal engineering, it is customary to calculate the probability of occurrence of extreme tidal sea levels by making the product of convolution of high tide densities with the survival function of storm surges. As 706 annual tides occur, the annual probability of exceeding a given level is chosen in reference to this figure of 1/706. In addition, because coastal structures are located at shallow depths, the conditions that require the structures in terms of their stability or wave overtoppings correspond to waves occurring with high water levels. That is

why the privileged situations are high tides. The analysed samples are therefore pairs of storm surge associated with wave height at high tide. However, for safety, the wave height that is chosen is the maximum value over the 3-hour time interval on either side of high tide. Kergadallan (2015) recommends selecting the maximum $H$ value within a time window centered on the time of high water. Using the same data as him, this recommendation is followed. Statistical samples therefore contain tuples of waves and storm surges values at high tides at the rate of 706 annual pairs. Separated by about 12 hours, the values

may not fully meet the independent and identically distributed (i.i.d) assumption. However, this aspect is not considered here as it is often accepted in practice (see for example Hawkes (2005)).

Another approximation is the assumption of the presence of a unique wave population. This assumption is also not completely valid when we consider the wave direction of extreme events. The topic has already been discussed by Hawkes (2002) and Mazas (2017, 2019), among others. The treatment of wave direction can also be considered as a fourth random variable of the oceanic forcing but has not been included in this work.

For low and moderate values the cumulative distribution functions (CDF) $F_H$, $F_T$ and $F_S$ are the empirical functions. For the strongest and extreme values, these cumulative distribution functions result from an adjustment of the exponential law. Survival functions, whether for storm surge or wave height, are therefore adjusted by piece by exponentially-type analytical functions as close as possible to empirical frequencies whereas extrapolations for extreme values use exponential laws.

## 2.2 Selection of the best bivariate copula by two methods

### 2.2.1 The error method

We illustrate the method for the random variables wave height $H$ and storm surge $S$. This method consists in determining the mean error $e$ between the calculated joint cumulative distribution function $F_{cal}(h, s, \theta)$ with the copula $C$, its parameter $\theta$ and the observed joint cumulative distribution function $F_{mes}(h, s)$

$$e = \frac{1}{n} \sum_{i=1,n} \left| \ln \frac{F_{cal}(h_i, s_i, \theta)}{F_{mes}(h_i, s_i)} \right| \tag{1}$$

with $n$ the number of pairs of values $(h_i, s_i)$.

For each copula, we first determine the parameter $\theta$ that minimizes the error $e$. We then select the copula with the lowest minimum mean error.

### 2.2.2 The maximum likelihood method

Let us call $X$ the sample of measures $(x_1, x_2, ..., x_n)$ with bivariate $x_i = (h_i, s_i)$, $i = 1, .., n$. The likelihood function $L$ is defined by equation (2)

$$L(X, \theta) = \prod_{i=1}^{n} f_{cal}(h_i, s_i, \theta) \tag{2}$$

where $f_{cal}$ is the probability density function of the bivariate cumulative distribution function $F_{cal}$. $\theta$ is the parameter of the copula.

The maximum likelihood method consists in finding the parameter $\theta$, which maximizes the probability of obtaining the sample (Tassi, 2004). Since likelihood is a product of density we take its log-likelihood in order to facilitate calculations. We can thus work with the sum and derive it with respect to $\theta$ as below

$$\frac{\partial}{\partial \theta} \ln L(X, \theta) = \frac{\partial}{\partial \theta} \ln \sum_{i=1}^{n} f_{cal}(h_i, s_i, \theta). \tag{3}$$

The best copula is the copula with the largest likelihood.

## 2.3 Construction of a trivariate copula

For more than two variables, $C$ is not generally a copula (impossibility theorem of Genest (1995)). According to Nelsen (2006), it is difficult to construct n-order copulas from n-1 copulas. We present two methods for the construction of trivariate copulas. In the first method, a trivariate copula generalizes the bivariate copula with three random variables and one parameter. In the second method, a trivariate copula associates two bivariate copulas with their two respective parameters.

### 2.3.1 Definition of a copula in dimension $d > 2$

A copula in dimension $d$ is a distribution function on $[0,1]^d$ whose marginal laws are uniform on $[0,1]$.

A copula is a function C: $[0,1]^d \longrightarrow [0,1]$, which satisfies the following three conditions

$$
\begin{aligned}
&i) \quad C(u_1, .., u_{i-1}, 0, u_{i+1}, .., u_d) = 0 \quad \forall u_i \in [0,1] \, ; \\
&ii) \quad\;\; C(1, \dots, 1, u_i, 1, \dots, 1) = u_i \qquad \forall u_i \in [0,1] \, ; \\
&iii) \quad\qquad C \text{ is } d - growing.
\end{aligned}
\tag{4}
$$

A function h : $[0,1]^d \longrightarrow$ R is called $d$-growing if for any hyper-rectangle $[a,b]$ of $R^d$, $V_h([a,b]) \geq 0$, where

$$
V_h([a,b]) = \Delta_a^b h(t) = \Delta_{a_d}^{b_d} \Delta_{a_{d-1}}^{b_{d-1}} \dots \dots \Delta_{a_2}^{b_2} \Delta_{a_1}^{b_1} h(t).
\tag{5}
$$

For each t, $\Delta_{a_i}^{b_i} h(t) = h(t_1, \dots, t_{i-1}, b_i, t_{i+1}, \dots, t_n) - h(t_1, \dots, t_{i-1}, a_i, t_{i+1}, \dots, t_n)$.

### 2.3.2 Trivariate copula with one parameter : a multi-level Archimedean trivariate

Since we are looking for the correlation between three variables, the first idea is to generalize the bivariate copula $C(u_1, u_2)$ to obtain $C(u_1, u_2, u_3)$. We must check that $C(u_1, u_2, u_3)$ is a copula, which is difficult. However Archimedean copulas like Gumbel and Clayton can be extended to an order greater than 2 using the property of Archimedean copulas (see appendix A). For a Clayton copula of order $n$, this gives

$$
C(u_1, \dots, u_n) \;=\; [u_1^{-\frac{1}{\theta}} + u_2^{-\frac{1}{\theta}} + \dots + u_n^{-\frac{1}{\theta}} - (\mathrm{n}-1)]^{-\theta}.
\tag{6}
$$

For Clayton copula of order 3, it gives

$$
C(u_1, u_2, u_3) \;=\; [u_1^{-\frac{1}{\theta}} + u_2^{-\frac{1}{\theta}} + u_3^{-\frac{1}{\theta}} - 2]^{-\theta}.
\tag{7}
$$

For Gumbel copula of order $n$, it gives

$$
C(u_1, \dots, u_n) = \exp\left(-\left[(-\mathrm{Ln}\,u_1)^\theta + (-\mathrm{Ln}\,u_2)^\theta + \dots + (-\mathrm{Ln}\,u_n)^\theta\right]^{\frac{1}{\theta}}\right) = \exp\left(-\left[\sum_i (-\mathrm{Ln}\,u_i)^\theta\right]^{\frac{1}{\theta}}\right).
\tag{8}
$$

For Gumbel copula of order 3, it gives

$$
C(u_1, u_2, u_3) = \exp\left(-\left[(-\mathrm{Ln}\,u_1)^\theta + (-\mathrm{Ln}\,u_2)^\theta + (-\mathrm{Ln}\,u_3)^\theta\right]^{\frac{1}{\theta}}\right).
\tag{9}
$$

By taking a single copula parameter for the three variables, we do not differentiate the pairwise correlations of the variables even though some variables may be more correlated than others.

### 2.3.3 Trivariate copula with two parameters : a fully nested hierarchical copula

To better take into account the correlations of variables two by two, one option is to build trivariate functions from bivariate copulas as a fully nested hierarchical copula as follows

$$
C(u_1, u_2, u_3) = C_1(C_2(u_1, u_2), u_3).
\tag{10}
$$

Corbella (2013) tests a fully nested hierarchical copula but he uses a unique bivariate copula and does not distinguish the two bivariate copulas $C_1$ and $C_2$. $C_1$ is a bivariate copula with $\theta_1$ as copula parameter. $C_2$ is a bivariate copula with $\theta_2$ as copula parameter. We must check that this function (10) is a copula and satisfies the properties of equations (4). We first aggregate

the two most correlated variables with the copula $C_2$ and its copula parameter. We then add the third random variable with the copula $C_1$ and its copula parameter. We will show later that this order provides the most robust copula.

### 2.3.4 Validity of copula properties for *2.3.3*

We do not know any general methods to build high order copulas from low order copulas (Durrleman, 2010). Generally $C(u_1, u_2, u_3) = C_1(C_2(u_1, u_2), u_3)$ is not a copula. To prove that $C(u_1, u_2, u_3)$ is a copula, we must check that $C(u_1, u_2, u_3)$

satisfies the three properties of equation (4) with $d = 3$, which is difficult. However Charpentier (2014) points out that $C$ is a copula if it satisfies i) or ii).

i) $C_1$ and $C_2$ are both Clayton or Gumbel copulas with parameters $\theta_1$ for $C_1$ and $\theta_2$ for $C_2$ positive and growing.

ii) $C_1$ and $C_2$ are both Archimedean copulas of respective generator $\phi_1$, $\phi_2$ with $\phi_2 \circ \phi_1^{-1}$ being the inverse of a Laplace transform.

For Gumbel and Clayton copulas $C_1$ and $C_2$ that are Archimedean copulas we check the condition (ii) that there is a function $f$ for which the inverse Laplace transform $T_L^{-1}$ satisfies

$$T_L^{-1}[f] = \phi_2 o \phi_1^{-1} \tag{11}$$

with $\phi_1$, $\phi_2$ generators of the copulas $C_1$ and $C_2$. $T_L[f](s) = \int_0^{+\infty} e^{-st} f(t)dt$ is the Laplace transform of $f$.

For $C_1$ and $C_2$ Clayton copulas we have as the generator of $C_2$ and as the inverse generator of $C_1$

$$\phi_2(t) = \frac{t^{-\theta_2}-1}{\theta_2} \; ; \phi_1^{-1}(t) = (1+\theta_1 t)^{-\frac{1}{\theta_1}}. \tag{12}$$

This gives

$$\phi_2 o \phi_1^{-1}(t) = \frac{[(1+\theta_1 t)^{\frac{\theta_2}{\theta_1}}-1]}{\theta_2}. \tag{13}$$

We can find that

$$T_L[\phi_2 o \phi_1^{-1}](s) = \frac{\left[e^{\frac{s}{\theta_1}}\Gamma(\frac{\theta_2}{\theta_1}+1,\frac{s}{\theta_1})-1\right]}{s\theta_2} \tag{14}$$

with $\Gamma(a,x)$ the incomplete Gamma function set by for a complex with real part$(a) > 0$

$$\Gamma(a,x) = \int_x^{+\infty} t^{a-1}e^{-t}\,dt. \tag{15}$$

We conclude that there is a function $f$ such that $\phi_2 o \phi_1^{-1} = T_L^{-1}[f]$ with

$$f = \frac{\left[e^{\frac{s}{\theta_1}}\Gamma(\frac{\theta_2}{\theta_1}+1,\frac{s}{\theta_1})-1\right]}{s\theta_2}. \tag{16}$$

For $C_1$ and $C_2$ Gumbel copulas we have as generator of $C_2$ and as the inverse generator of $C_1$

$$\phi_2(t) = (-\ln t)^{\theta_2} \; ; \phi_1^{-1}(t) = e^{-t^{\frac{1}{\theta_1}}}. \tag{17}$$

This gives

$$\phi_2 o \phi_1^{-1}(t) = \left[-\ln\left(e^{-t^{\frac{1}{\theta_1}}}\right)\right]^{\theta_2} = t^{\frac{\theta_2}{\theta_1}}. \tag{18}$$

We can find that

$$T_L[\phi_2 o \phi_1^{-1}](s) = T_L\left(t^{\frac{\theta_2}{\theta_1}}\right) = \Gamma\left(\frac{\theta_2}{\theta_1}\right)s^{-\frac{\theta_2+\theta_1}{\theta_1}} \tag{19}$$

with $\Gamma$ Gamma function defined by :

$$\Gamma(a) = \int_0^{+\infty} y^{a-1}e^{-y}\,dy. \tag{20}$$

We conclude that there is a function $f$ such that $\phi_2 o \phi_1^{-1} = T_L^{-1}[f]$ with

$$f = \Gamma\left(\frac{\theta_2}{\theta_1}\right)s^{-\frac{\theta_2+\theta_1}{\theta_1}}. \tag{21}$$


## 2.4 Determination of the contour of equal joint exceedance probability

The determination of the contour of equal joint exceedance probability $P(H > h, T > t, S > s)$ consists in obtaining all the variables $(H, T, S)$ associated with different return periods : $T_{10}$ (10-year event), $T_{100}$ (100-year event) and $T_{1000}$ (1000-year

event).

### 2.4.1 Bivariate probability without tide

We deal with a set of pairs of values $(h, s)$ that satisfy

$$\bar{C}[\bar{F}_H, \bar{F}_s] = f_{10}, f_{100} \text{ or } f_{1000}. \tag{22}$$

$\bar{C}$ is the selected bivariate survival copula. $\bar{F}_H, \bar{F}_s$ are survival functions associated with the variables. The values $f_{10}, f_{100}$ or $f_{1000}$ are the frequencies corresponding to the ten-year, hundred-year and thousand-year periods i.e 1/7060, 1/70600 and 1/706000.

### 2.4.2 Bivariate probability with tide

The bivariate probability with tide requires the development of the copula connecting wave height and storm surge. We can then define the joint survival function of the wave height and the storm surge. The chosen calculation method favors high tide. The sea levels considered are therefore the sums of the astronomical high tide (generated by the attraction of the moon and the sun without weather disturbance) and the storm surges raised at the time of these astronomical high tides. This method is of course valid only for macrotidal seas. The probability that the sea level at high tide $N$ exceeds a given value $n$ is expressed as follows

$$P(n) = P[N > n] = \int_{M_{min}}^{M_{max}} f_M(z)\bar{F}_S(n - z)\mathrm{d}z \tag{23}$$

with $z$ the height of the high tide, between the minimum and maximum values $M_{min}$ and $M_{max}$ respectively at high tide, $f_M(z)dz$ the probability that the high tide is between $z$ and $z + \mathrm{d}z$ and $\bar{F}_S(s)$ the probability of observing a storm surge $S$ larger than s, thus $\bar{F}_S(s) = P(S > s)$. The equation (23) is established by Simon (1994).

The bivariate survival function for wave height $H$ and sea level $N$ is therefore written as follows

$$\bar{F}_{HN}(h, n) = \int_{M_{min}}^{M_{max}} f_M(z)\bar{F}_{HS}(h, n - z)dz. \tag{24}$$

Introducing the survival copula $\bar{C}$, the final equation is

$$\bar{F}_{HN}(H, N) = \int_{M_{min}}^{M_{max}} f_M(z)\bar{C}(\bar{F}_H(h), \bar{F}_S(n - z))dz. \tag{25}$$

The set of pairs ($h$, $n$) corresponding to the different return periods, the ten-year, hundred-year and thousand-year periods, satisfies

$$\int_{M_{min}}^{M_{max}} f_M(z)\bar{C}(\bar{F}_H(h), \bar{F}_S(n - z))dz = f_{10}, f_{100} \text{ or } f_{1000}. \tag{26}$$

It is thus possible to represent the contour of equal joint exceedance probability associated with the variables wave height and sea level.

### 2.4.3 Trivariate probability without tide

Here we have chosen the method of construction of a trivariate copula with two parameters known as fully nested hierarchical copula. We have

$$\bar{F}_{HT}(h, t) = \bar{C}_1(\bar{F}_H(h), \bar{F}_T(t)), \tag{27}$$

$$\bar{F}_{HTS}(h, t, s) = \bar{C}_2(\bar{F}_{HT}(h, t), \bar{F}_S(s)), \tag{28}$$

with $\bar{C}_1$ and $\bar{C}_2$ the selected bivariate survival copula. From equations (27) and (28) we therefore obtain the equation that follows

$$\bar{F}_{HTS}(h, t, s) = \bar{C}_2\left(\bar{C}_1(\bar{F}_H(h), \bar{F}_T(t)), \bar{F}_S(s)\right). \tag{29}$$

The triplets of values ($h$, $t$, $s$) corresponding to the different return periods, $T_{10}$ (10-year event), $T_{100}$ (100-year event) and $T_{1000}$ (1000-year event) satisfy

$$\bar{C}_2\left(\bar{C}_1(\bar{F}_H(h), \bar{F}_T(t)), \bar{F}_S(s)\right) = f_{10}, f_{100} \text{ or } f_{1000}. \tag{30}$$

It is thus possible to represent the contours of equal joint exceedance probability associated with the variables wave height, wave period and sea level.

### 2.4.4 Trivariate joint exceedance probability with tide

The trivariate survival function for wave height $H$, wave period $T$ and sea level $N$ is written as follows

$$\bar{F}_{HTN}(h,t,n) = \int_{M_{min}}^{M_{max}} f_M(z)\bar{F}_{HTS}(h,t,n-z)dz. \tag{31}$$

This can be written by introducing the selected survival copula $\bar{C}_2$

$$\bar{F}_{HTN}(h,t,n) = \int_{M_{min}}^{M_{max}} f_M(z)\,\bar{C}_2(\bar{F}_{H,T}(h,t),\bar{F}_S(n-z))dz. \tag{32}$$

Introducing the survival copula $\bar{C}_1$ connecting $\bar{F}_H$ and $\bar{F}_T$, the final equation is

$$\bar{F}_{HTN}(h,t,n) = \int_{M_{min}}^{M_{max}} f_M(z)\,\bar{C}_2\left(\bar{C}_1(\bar{F}_H(h),\bar{F}_T(t)),\bar{F}_S(n-z)\right)dz. \tag{33}$$

The triplets of values ($h$, $t$, $n$) corresponding to the different return periods, $T_{10}$ (10-year event), $T_{100}$ (100-year event) and $T_{1000}$ (1000-year event) satisfy

$$\int_{M_{min}}^{M_{max}} f_M(z)\,\bar{C}_2\left(\bar{C}_1(\bar{F}_H(h),\bar{F}_T(t)),\bar{F}_S(n-z)\right)dz = f_{10}, f_{100} \text{ or } f_{1000}. \tag{34}$$

It is thus possible to represent the contours of equal joint exceedance probability associated with the variables wave height, wave period and sea level with tide.

**2.5 Tail dependence of the sample**

It is necessary to treat the extreme events that are characterized by a very low occurrence. The difficulty of taking them into account is of a statistical nature: the scarcity of observations. In order to take the extreme events into account, we introduce the concept of tail dependence. For a bivariate copula, the tail dependence measures the probability of simultaneous extreme realizations (Clauss, 2009). It is a highly relevant tool for the study of extreme values. We distinguish lower and upper tail dependences. They are characterized by their lower and upper tail dependence coefficients that are deduced from the following conditional probabilities, whose value is given by equations (35) and (36) that are given by (Clauss, 2009) for cumulative distribution functions $U_1$ and $U_2$

$$P(U_1 \le u_1 | U_2 \le u_2) = \frac{P(U_1 \le u_1, U_2 \le u_2)}{P(U_2 \le u_2)} = \frac{C(u_1, u_2)}{u_2}, \tag{35}$$

$$P(U_1 > u_1 | U_2 > u_2) = \frac{P(U_1 > u_1, U_2 > u_2)}{P(U_2 > u_2)} = \frac{1 + C(u_1, u_2) - u_1 - u_2}{1 - u_2}. \tag{36}$$

Since we fix the lower tail dependence coefficient $\lambda_L$ and upper tail dependence coefficient $\lambda_U$ by equations

$$\lambda_L = lim_{u \to 0}\, P(U_1 \le u | U_2 \le u), \tag{37}$$

$$\lambda_U = lim_{u \to 1}\, P(U_1 > u | U_2 > u), \tag{38}$$

we deduce the definitions of tail dependence coefficients.

**Definition:** The lower tail dependence coefficient is defined by

$$\lambda_L = \ lim_{u \to 0} \frac{C(u,u)}{u}. \tag{39}$$

The copula $C$ has a lower tail dependence if $\lambda_L$ exists with $\lambda_L \in\ ]0,1]$.

If $\lambda_L = 0$ then the copula does not have a lower tail dependence.

**Definition :** The upper tail dependence coefficient is defined by

$$\lambda_U = lim_{u \to 1} \frac{1 + C(u,u) - 2u}{1 - u}. \tag{40}$$

The copula $C$ has an upper tail dependence if $\lambda_U$ exists with $\lambda_U \in\ ]0,1]$.

If $\lambda_U = 0$ then the copula does not have an upper tail dependence.

In the following section, we use survival copula $\bar{C}$ and survival function $\bar{u}$. The lower tail dependence corresponds then to high wave heights and water levels.

The tail dependences of the different copulas are determined in (Nelsen, 2006) and (Roncalli, 2002) from their tail dependence coefficients. They are expressed in Table 1. Ali-Mikhail-Haq copula and Gaussian copula will be noted AMH and Gauss respectively in what follows.

| Copula | $\lambda_L$ | $\lambda_U$ |
|---|---|---|
| AMH | 0 | 0 |
| Clayton | $2^{-\frac{1}{\theta}}$ | 0 |
| Frank | 0 | 0 |
| Galambos | 0 | $2^{-\frac{1}{\theta}}$ |
| Gauss | 0 | 0 |
| Gumbel | 0 | $2 - 2^{\frac{1}{\theta}}$ |
| Survival Gumbel | $2 - 2^{\frac{1}{\theta}}$ | 0 |
| Joe | 0 | $2 - 2^{\frac{1}{\theta}}$ |
| Plackett | 0 | 0 |
| Student | $2t_{v+1}\left(-\sqrt{v+1}\sqrt{\frac{1-\theta}{1+\theta}}\right)$ | $2t_{v+1}\left(-\sqrt{v+1}\sqrt{\frac{1-\theta}{1+\theta}}\right)$ |

**Table 1** : Tail dependence coefficients.

We find that some copulas do not have lower and upper tail dependence coefficients. They are inappropriate in case of extreme dependence. Some copulas have a lower tail dependence, others have an upper tail dependence. The tail dependence of the sample is firstly checked. For this we graphically represent the evolution of $\frac{\bar{C}(\bar{u},\bar{u})}{\bar{u}}$ and determine its limit when $\bar{u}$ tends to 0. Since $\bar{u}$ is used, we note the tail dependence coefficients $\bar{\lambda}_L$ and $\bar{\lambda}_U$. We can therefore decide whether the sample has or has not a lower or upper tail dependence. In choosing the copula, it is essential to satisfy the class of tail dependence of the sample.

If the sample does not have a tail dependence, then the use of Gaussian copula or other copula with the same class od tail dependence is recommended. If the sample has a lower tail dependence, the use of a copula with a lower tail dependence or the survival copula with an upper tail dependence is recommended. If the sample has an upper tail dependence, the use of a copula with an upper tail dependence or the survival copula with a lower tail dependence is recommended. We can also deduce the parameter of the copula from the tail dependence coefficient given by the sample. The method that is proposed here for

assessing the sample dependence refers to lower tail dependence. Other methods exist such as the chi-plot proposed by Fisher and Switzer (1985, 2001) and used in coastal analyses by Mazas (2017) for instance.

**3 Results for bivariate copulas**

We select the most appropriate copulas at both the Le Havre and Saint-Malo (Northern France) sites using two methods. We analyze the class of tail dependence of the two samples. We represent the contour of equal joint exceedance probability with

280 the selected copulas for three return periods in order to assess the relevance of the copulas.

**3.1 Statistical law for adjusting wave height, wave period and storm surge**

The representation of the contours requires knowledge of the statistical laws of adjustment of the different parameters. We therefore present these laws. For the two sites of Saint Malo and Le Havre we have used data files that provide the values for wave height, wave period and storm surge at high tide over a time period of about twenty years. The file for Le Havre site

includes, for example, around 15.000 values. The wave data are extracted from the Anemoc numerical database. Sea levels at high tide are extracted from tide gauge measurements. The astronomical tide is obtained from the Shom Predit software. Adjustments of the statistical laws are made according to the POT method on the basis of the exponential law.

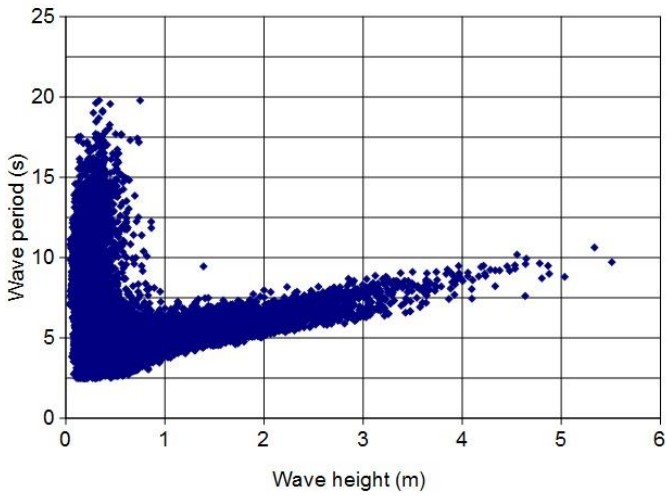

**Figure 2 :** Set of wave data in Le Havre (1979 – 2002).

The copula parameters are calibrated from samples where wave height values less than one meter are excluded (see Figure 2), thus reducing the sample size to about 3.000 values. The copulas are fitted to all pairs/triplets of observations where the wave height exceeds one meter. This threshold of one meter that is used for filtering wave height excludes the swells and leaves only a very homogeneous population of pure wind waves. This treatment removes long wave periods and increases the dependence between wave height and wave period.

**3.2 Current pratice : Defra method**

The use of the simplified Defra method in Ciria *et al.* (2007) is common among European coastal engineers for the study of wave overtopping or armor damages in coastal structures. It refers to the complete Defra method presented for example by Hawkes (2005) that is based on the Gauss copula. The complete Defra method is close to the method that is used in this paper. The main difference is the choice of the Gauss copula that does not present tail dependence. The simplified Defra method refers to univariate survival functions $\bar{F}_H$ and $\bar{F}_S$ rather than cumulative distribution functions of wave height and storm surge as coastal engineers usually work with exceedance probability rather than with non-exceedance probability. In this simplified method, the bivariate survival function is related to univariate survival functions by expression

$$\bar{F}_{HS} = \text{FD}\ \bar{F}_H \bar{F}_S. \tag{41}$$

In France, the order of magnitude for the dependence factor FD coefficient is about 20. Kergadallan (2013) recommends however a minimum value of 25 for safety reasons. This factor corresponds to a weak dependence. For a very strong dependence, FD is between 500 and 1000.

The bivariate survival functions $\bar{F}_{HS}$ of table 4.15 of Rock Manual (Ciria *et al.*, 2007) are determined with equation (41). Figure 3 shows the differences between observed bivariate survival functions and calculated bivariate survival functions using the simplified Defra method. The points of calculations in blue lie far from the first bisector in black in the figure. This shows that the use of the Defra simplified method is inappropriate. This is due mostly to the use of the simplified Defra method of equation (41) but the complete Defra method with Gauss copula would not represent also perfectly the extreme events because Gauss copula has not tail dependence as we will see later.

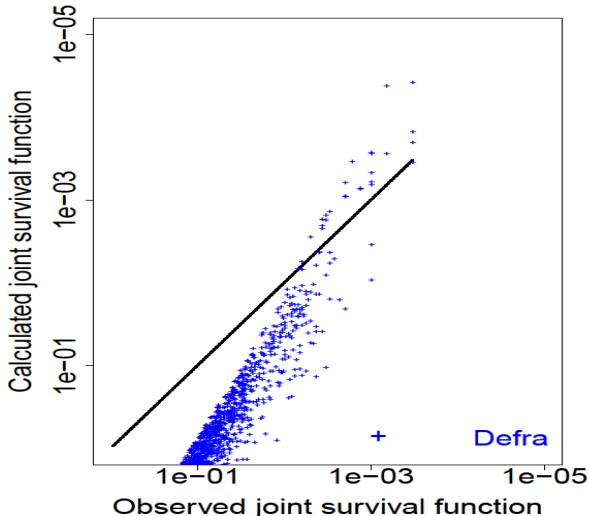

**Figure 3 :** Comparison of calculated (with Defra method) and observed joint frequency for Le Havre.


In order to improve the results we now introduce the copula theory.

### 3.3 Analysis of the tail dependence

The sample is analyzed in order to determine its tail dependence. Indeed, the result will condition the choice of the copula depending on whether the sample has the same class of tail dependence as the copula or not. In equations (22), (26), (30), (34)

the survival copula $\bar{C}$ is used with survival functions. In the following section, we use survival copula $\bar{C}$ and survival function $\bar{u}$. Upper tail dependence and lower tail dependence will be inverted. We are interested in the extreme events with high wave heights and water levels. For survival copula $\bar{C}$, we determine below its limit for survival function $\bar{u}$ tending to 0. The lower tail dependence corresponds to these high wave heights and water levels.

a)   Saint-Malo                                        b)   Le Havre

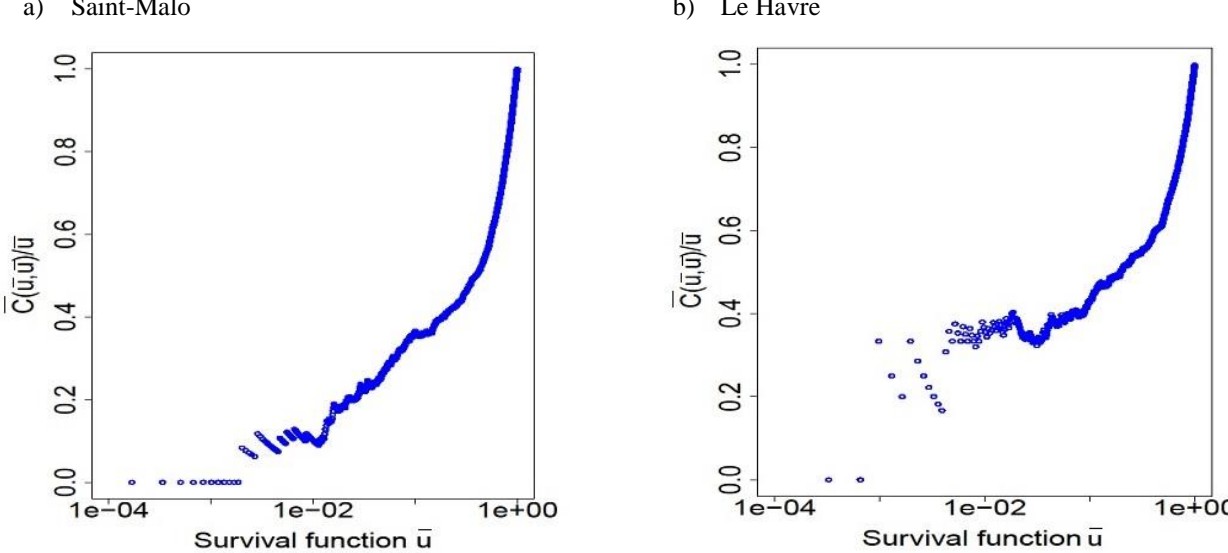

**Figure 4 :** $\frac{\bar{C}(\bar{u},\bar{u})}{\bar{u}}$ for a) Saint-Malo and b) Le Havre samples.

For the Saint-Malo sample, $\frac{\bar{C}(\bar{u},\bar{u})}{\bar{u}}$ tends to around 0.2 when $\bar{u}$ tends to 0.

For the Le Havre sample, $\frac{\bar{C}(\bar{u},\bar{u})}{\bar{u}}$ tends to around 0.4 when $\bar{u}$ tends to 0.

Using the survival function $\bar{u}$ , these two samples have a lower tail dependence which justifies the use of the Clayton copula. We determine the Clayton copula parameter from the lower tail dependence coefficient of the sample. With the Clayton copula,

we can determine the value of its copula parameter in Saint-Malo and Le Havre with equation

$$\theta = -\frac{\ln 2}{\ln \bar{\lambda}_L}. \qquad (42)$$

This copula parameter is 0.43 and 0.76 respectively.

Note : as the Gumbel copula has an upper tail dependence it is not recommended. On the contrary, its survival copula with a lower tail dependence is appropriate. This analysis of the sample makes it possible to understand why the survival Gumbel copula gives a minimum of error close to the minimum error of the Clayton copula. We can therefore expect survival Gumbel
copula results to be close to the results obtained by Clayton copula.

### 3.4 Selection of the best bivariate copula for Le Havre and Saint-Malo samples

### 3.4.1 The log-likelihood method

| Copula Sites | Copula Parameter $\theta$ Saint-Malo | Copula Parameter $\theta$ Le Havre | Maximum likelihood Saint-Malo | Maximum likelihood Le Havre |
|---|---|---|---|---|
| Ali-Mikhail-Haq | 0.71 | 0.96 | 196 | **375** |
| Clayton | 0.38 | 0.74 | **291** | **387** |
| Franck | 1.25 | 2.67 | 124 | 271 |
| Galambos | 0.31 | 0.54 | 41 | 175 |
| Gauss | 0.22 | 0.42 | 149 | 297 |
| Gumbel | 1.09 | 1.29 | 52 | 185 |
| Survival Gumbel | 1.18 | 1.39 | **243** | **372** |
| Joe | 1.03 | 1.21 | 4 | 76 |
| Plackett | 1.88 | 3.58 | 127 | 277 |
| Student | 0.22 | 0.42 | **157** | **303** |

**Table 2** : Copula parameter and maximum likelihood for the different survival copulas in Saint-Malo and Le Havre.

For the set of survival copulas we determine their maximum likelihood with their parameter. We will select the survival copula with the largest likelihood among those which possess the same class of tail dependence as the sample with the largest likelihood. In bold in Table 2 are presented the survival copulas with a lower tail dependence: Clayton, survival Gumbel and Student. AMH is added in bold when copula parameter is close to 1. We will come back later to this special property of AMH copula. The Gauss copula has a relatively large likelihood. However, it does not have a tail dependence and cannot therefore
correctly represent the tail dependence.

For the Saint-Malo sample, we choose the Clayton copula, which has the same class of tail dependence as the sample, with a log-likelihood of 291 in table 2. For the Le Havre sample, we also choose the Clayton copula, which has the same class of tail dependence as the sample, with a log-likelihood of 387.

The Clayton copula parameters obtained by the tail dependence coefficients come close to those obtained by the log-likelihood
method for the Le Havre sample (3.040 values) and the Saint-Malo sample (5.888 values).

For Saint-Malo, we obtain as 0.38 the parameter of the Clayton copula using the method of maximum likelihood and 0.43 with the tail dependence coefficient.

For Le Havre, we obtain 0.74 as the parameter of the Clayton copula using the method of maximum likelihood and 0.76 with the tail dependence coefficient.

Even if this comparison is satisfactory, the method can be sensitive to the data and the way to determine the limit. Caillault and Guegan (2005) propose for example two methods which allow to estimate the copula characterizing the bivariate distribution function of a pair of markets. One method privileges the extreme behavior of the bivariate distribution function of the pair and the second one is based on the estimates of the copulas' parameter using a pseudo log-likelihood method. They conclude that the two approaches give different estimates of the tail dependence.


The value of the log-likelihood of the survival Gumbel copula is approximately as large as the log-likelihood of the Clayton copula. In addition, the survival Gumbel copula has the same class of tail dependence as the Clayton. It is therefore potentially as suitable as the Clayton copula.

### 3.4.2 The error method for the Clayton, Gumbel and survival Gumbel Copula

In order to select the most relevant copula, we represent the mean error $e$ between the calculated survival function $F_{cal}(h, s, \theta)$ with the copula $C$ and its parameter and the measured $F_{mes}(h, s)$.

a) Saint-Malo                                    b)    Le Havre

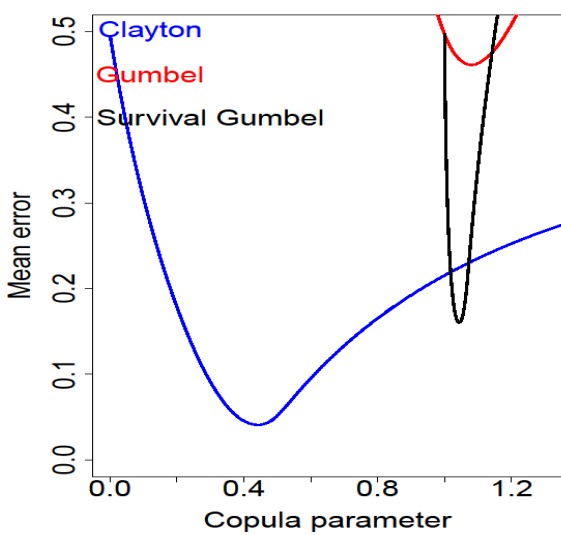
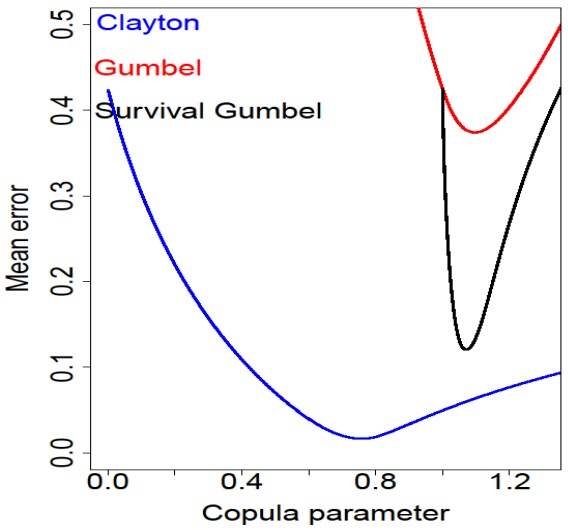

**Figure 5 :** Evolution of the error according to the Clayton, Gumbel and survival Gumbel copula parameter in a) Saint-Malo and b) Le Havre.

Figure 5 for the ports of Saint-Malo and Le Havre shows that the error that is obtained with the survival Gumbel copula is very
close to that obtained with the Clayton copula. The curve of the error obtained by the survival Gumbel copula however has a very acute minimum. Obtaining the parameter of this copula will therefore be very sensitive to the value of its minimum error. It will therefore be necessary to determine it very precisely.

Note: Gumbel and Clayton copula parameter supports are different and are $[1, +\infty[$ and $]0, +\infty[$ respectively.

We note Emin the minimum of the mean error e and Error rate = exp (Emin) - 1. Table 3 below shows the results obtained for
Saint Malo and Le Havre.

| Copula | Emin | Emin | Error rate | Error rate | Parameter | Parameter |
|---|---|---|---|---|---|---|
| Sites | Saint-Malo | Le Havre | Saint-Malo | Le Havre | Saint-Malo | Le Havre |
| **Gumbel** | 0.45 | 0.37 | 57 % | 44 % | 1.03 | 1.10 |
| **Survival Gumbel** | 0.18 | 0.12 | 20 % | 13 % | 1.02 | 1.07 |
| **Clayton** | 0.05 | 0.03 | 5 % | 3 % | 0.40 | 0.76 |

**Table 3** : Emin, error rate and copula parameter for the for the different survival copulas in the ports of Le Havre and Saint Malo : Clayton, Gumbel and survival Gumbel.

Table 3 is used to verify that Clayton copula is the most robust copula. It also appears that survival Gumbel copula is also an
appropriate option.

We have therefore shown by two methods that the Clayton copula is the most relevant for the Saint-Malo and Le Havre sites. The parameters of the copula obtained by the error method are close to those obtained by the method of maximum likelihood for the Clayton copula.

### 3.5 Comparison of observed and calculated joint frequencies

In order to assess the robustness of the copulas, we show in Figure 6 the observed and calculated joint frequencies for the Le Havre sample (3.040 pairs of values). The copula represents reality more closely when the points approach the bisector $y = x$. The simplified Defra method currently in use does not give a good representation of the reality of the joint frequencies for wave height and storm surge. The Clayton copula provides a good representation. In contrast, the Gumbel copula gives a bad representation. The explanation is in the analysis of the sample carried out in section 3.3: we showed that the sample had a lower tail dependence whereas the Gumbel copula has an upper tail dependence. The survival Gumbel copula provides on the contrary a good representation of the reality of joint frequencies for wave height and storm surge. The explanation lies in the fact of introducing the survival copula. The tail dependence of the survival Gumbel copula is opposite to the tail dependence of the Gumbel copula. The survival Gumbel copula has therefore the right class of tail dependence. The results obtained by AMH copula are surprisingly correct. Kumar (2010) shows that the AMH copula does not have tail dependence except if the copula parameter is equal to 1. In our case, the copula parameter is close to 1. The copula seems therefore to behave like a copula with a lower tail dependence. We show here the utility of the Clayton copula in comparison with the Gumbel copula and the Defra method that is currently in use.

The results highlight the importance in copula selection of the class of tail dependence of the sample. If the sample has a tail dependence it is necessary to select a copula with the same tail dependence. The Clayton copula that has the same class of tail dependence as the sample gives a calculated joint frequency close to the observed joint frequency. Conversely the Gumbel copula does not correctly represent the observed joint frequency: it moves away from the bisector for the extreme points. This is because the sample has a tail dependence opposite to that of the Gumbel copula. In order to restore the proper tail dependence, we resort to the survival copula. The latter comes close the bisector but is slightly less robust than the Clayton copula. It should be noted that calibration is performed on the entire sample. By truncating the sample for joint frequency values below 0.01, we would have obtained a much larger parameter for the Gumbel copula with results that are closer to measurements.

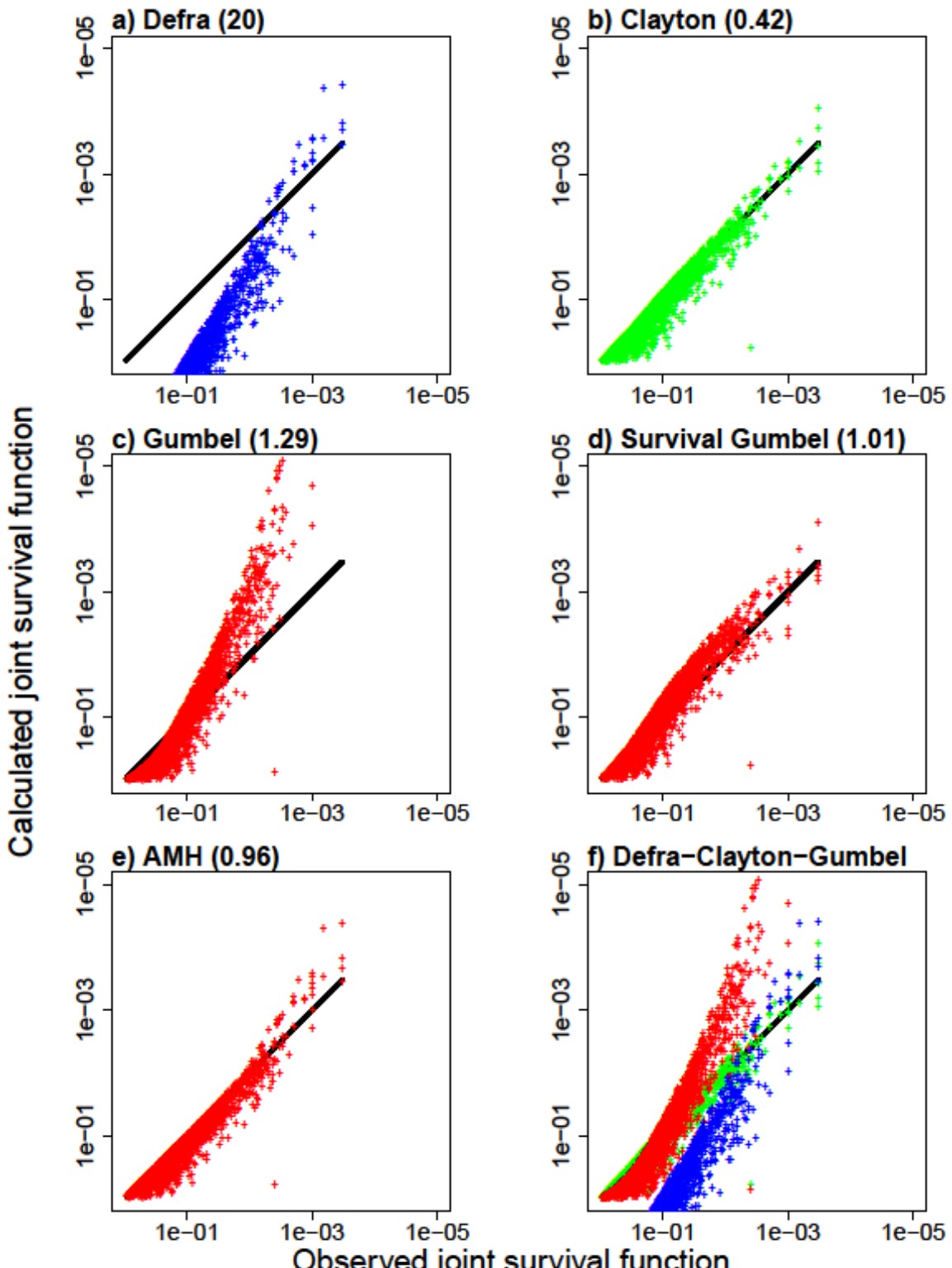

**Figure 6 :** Comparison of the observed joint survival and the calculated joint survival function for Le Havre with (a) Defra method, (b) Clayton (0.42), (c) Gumbel (1.29), (d) Survival Gumbel (1.01), (e) AMH (0.96) and (f) Defra-Clayton-Gumbel.

**3.6 Contours of equal joint exceedance probability with bivariate copula**

**3.6.1 Contours without tide for the Clayton, Gumbel, and survival Gumbel copulas and the Defra method**

Figure 7 shows the joint exceedance probability ($H$, $S$) for the Le Havre (3.040 values) samples respectively with Clayton, Gumbel, survival Gumbel copulas and the Defra method.

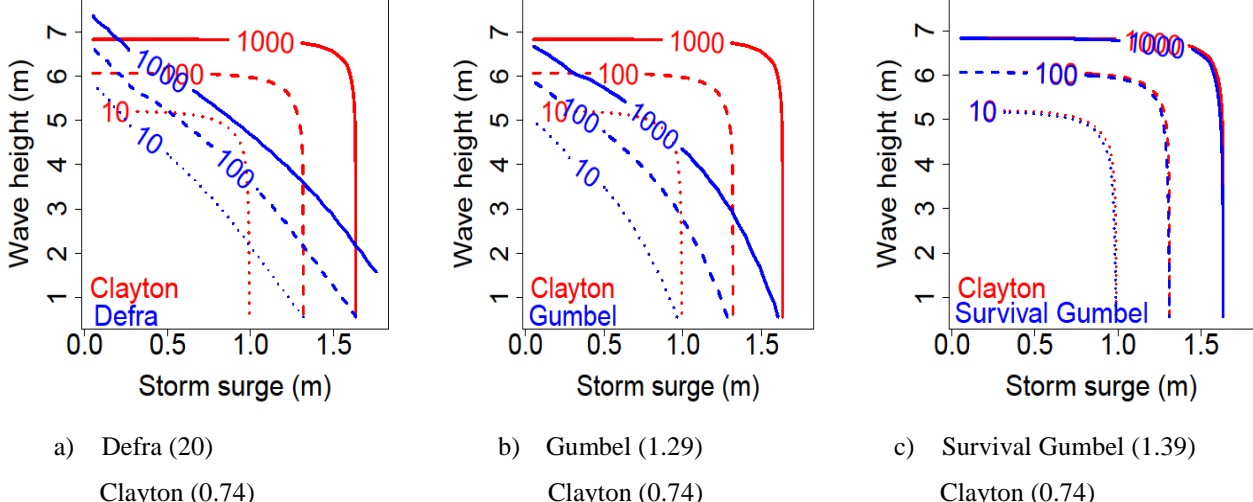

a)  Defra (20)         b)  Gumbel (1.29)         c)  Survival Gumbel (1.39)

Clayton (0.74)          Clayton (0.74)          Clayton (0.74)

**Figure 7** : Contours of equal joint exceedance probability with Clayton (0.74), Defra (20), Gumbel (1.29) and survival Gumbel (1.39) for return periods of 10, 100 and 1000 years for Le Havre.

Figures 7a, 7b and 7c present the comparison of Clayton with respectively Defra, Gumbel and survival Gumbel. Contours of

equal joint exceedance probabilities obtained by Clayton are very far from those obtained by Gumbel and the Defra method. On the contrary, the joint exceedance curves obtained using the survival Gumbel copula are very similar to those obtained with Clayton. Results are therefore very sensitive to the choice of copula. A poor choice may lead to undersizing and may have economic consequences.

**3.6.2 Contours with tide for Clayton copula**

Figure 8 shows the contours of equal joint exceedance probability respectively for the port of Saint-Malo (5.000 tidal values) and the Le Havre sample (22.000 tidal values) with the Clayton copula.

a)  Saint-Malo                                        b)  Le Havre

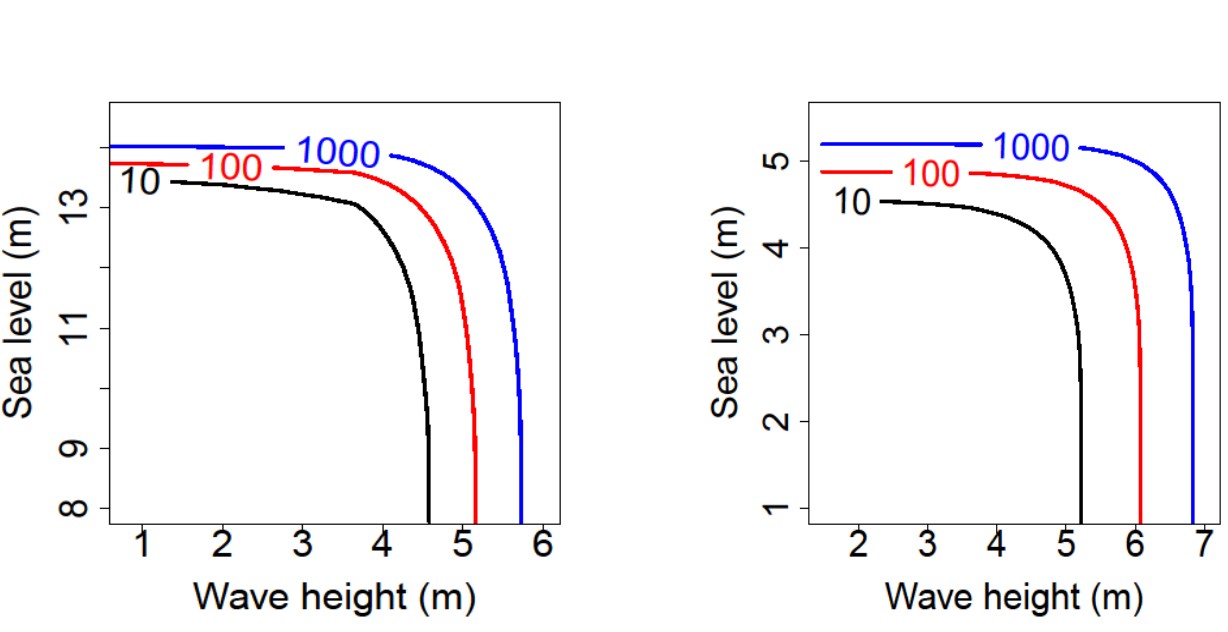

Figure 8 : Joint exceedance probability obtained with a) Clayton copula (0.38) for Saint Malo b) with Clayton copula (0.74) for Le Havre with tide for return periods of 10, 100 and 1000 years.

With tide the effect of storm surge on the sea level is small. The tidal range, which has an amplitude much larger than the storm surge especially for the port of Saint Malo, mitigates the variations due to the storm surge. In particular, for the port of

Saint-Malo, it can be seen that sea level is less sensitive to variations in the return periods than storm surge (cf. Figure 8).

**3.7 Conclusion on selecting of the best bivariate copula**

We selected the Clayton copula for the ports of Le Havre and Saint-Malo using three methods. In order to validate the Clayton copula, we analyzed samples from 19 sites of the French coast along the Atlantic and English Channel with the maximum likelihood method. We always obtained the greatest maximum likelihood with the Clayton copula or the AMH copula (see

appendix C). The sample always has lower tail dependence (see appendix B). Even though in some sites the AMH copula provides a larger likelihood than the Clayton copula, it should not be chosen because it has a particular kind of behavior. It has a lower tail dependence if the copula parameter is 1 (or close to 1 in practice). If the parameter is not 1, the AMH copula does not have tail dependence and its interests disappears. Since the robustness depends on the copula parameter and on the site, it cannot be recommended for a general use. We can therefore conclude that the Clayton copula is the most appropriate copula

for our application. For this purpose, the Table 4 gives the parameters of the different sites.

| Sites | Parameter |
|---|---|
| Dunkerque | 0.67 |
| Calais | 0.56 |
| Boulogne-sur-mer | 0.77 |
| Dieppe | 0.80 |
| Le Havre | 0.95 |
| Cherbourg | 0.49 |
| Saint-Malo | 0.48 |
| Roscoff | 0.41 |
| Le Conquet | 0.54 |
| Brest | 0.55 |
| Concarneau | 0.93 |
| Port-Tudy | 0.92 |
| Saint-Nazaire | 1.05 |
| Saint-Gildas | 0.90 |
| La Rochelle | 1.00 |
| Port-Bloc | 0.95 |
| Bayonne | 0.43 |
| Socoa | 0.43 |

**Table 4** : Clayton parameters for the different sites.

We show in Figure 9 that there is a coastal area with a maximal dependence from Concarneau to Port-Bloc (in grey in the figure). This area is the most exposed to wind that comes mainly from the West direction along the French Atlantic Coast.

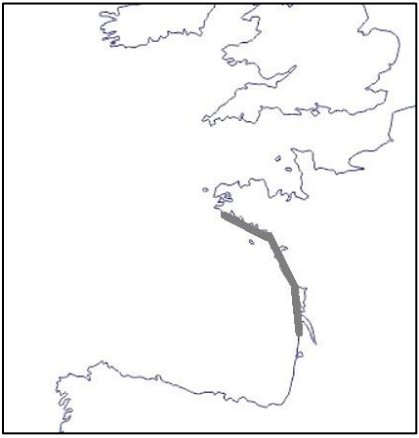

**Figure 9 :** The coast with the maximal dependence from Concarneau to Port-Bloc.

# 4 Results for trivariate copulas

## 4.1 Methodology

We have tested hierarchical construction using a fully nested hierarchical Archimedean copula. In this type of construction, we build a bivariate copula between two variables, then we create a trivariate copula with the previous copula and the third variable using another bivariate copula. Unlike Corbella (2013), who uses the same copula parameter for the two bivariate copulas, we introduce two different copula parameters.

## 4.2 Construction of the best trivariate copula for the port of Le Havre

With the fully nested hierarchical copula method, we first determine the most appropriate copula for two parameters: $(T, S)$, $(H, T)$ and then $(H, S)$. We construct the bivariate distribution function using the selected copula for the two most correlated variables. We determine the most relevant copula between the function obtained with the two most correlated variables and the third variable.

### 4.2.1 Bivariate copula for the three random variables

To determine the best bivariate copula we assess the maximum likelihood between $(\bar{F}_H, \bar{F}_S)$, $(\bar{F}_T, \bar{F}_S)$ and $(\bar{F}_H, \bar{F}_T)$ with the different copulas in Table 5.

For all three combinations, the Clayton copula and survival Gumbel copula still have the largest maximum likelihood value. In addition, we find that for the combination $(H, T)$ the log-likelihood is significantly higher. This result is related to the fact that the parameters $(H, T)$ are the most correlated parameters all the more since we deal with a very homogeneous population of pure wind waves as it is noticed in section 3.1.

We can write

$$\bar{F}_{H,T} = [(\bar{F}_H)^{-2.37} + (\bar{F}_T)^{-2.37} - 1]^{-\frac{1}{2.37}}. \tag{43}$$

| Copula | Parameter | Parameter | Parameter | Maximum likelihood | Maximum likelihood | Maximum likelihood |
|---|---|---|---|---|---|---|
| | (H,S) | (T,S) | (H,T) | (H,S) | (T,S) | (H,T) |
| Gumbel | 1.29 | 1.18 | 1.99 | 185 | 82 | 1059 |
| Survival Gumbel | 1.39 | 1.25 | 2.37 | 372 | 205 | 1584 |
| Clayton | **0.73** | **0.50** | **2.37** | **387** | **221** | **1565** |
| Gauss | 0.42 | 0.31 | 0.77 | 296 | 149 | 1369 |
| Franck | 0.67 | 1.83 | 7.27 | 271 | 139 | 1333 |
| Student | 0.42 | 0.30 | 0.77 | 303 | 159 | 1404 |
| Plackett | 3.58 | 2.49 | 15.64 | 277 | 138 | 1349 |
| Joe | 1.26 | 1.14 | 2.06 | 76 | 26 | 651 |
| Galambos | 0.83 | 0.61 | 1.25 | 175 | 75 | 1038 |

**Table 5** : Log-likelihood and copula parameter for the different bivariate copulas between the parameters $H$ and $S$, $T$ and $S$ then $H$ and $T$.

### 4.2.2 Determination of the best trivariate copula

We determine the maximum likelihood between $\bar{F}_{H,T}$ and $\bar{F}_S$ with the different copulas in Table 6.

We obtain the largest log-likelihood for Clayton copula, with a parameter of 0.56, which gives

$$\bar{F}_{H,T,S} = \left[(\bar{F}_{H,T})^{-0.56} + (\bar{F}_S)^{-0.56} - 1\right]^{-\frac{1}{0.56}}. \tag{44}$$

In conclusion, we have thus aggregated the most correlated $H$ and $T$ parameters with the best performing Clayton copula. We also used Clayton copula to aggregate $\bar{F}_{H,T}$ and $\bar{F}_S$. The aggregation requires two different parameters.

| Copula | Parameter | Maximum likelihood |
|---|---|---|
| Gumbel | 1.25 | 120 |
| Survival Gumbel | 1.29 | 263 |
| Clayton | **0.56** | **289** |
| Gauss | 0.36 | 195 |
| Franck | 2.08 | 156 |
| Student | 0.35 | 215 |
| Plackett | 2.84 | 165 |
| Joe | 1.72 | 35 |
| Galambos | 0.50 | 111 |

**Table 6** : Log-likelihood and copula parameter for different bivariate copulas between $\bar{F}_{H,T}$ and $\bar{F}_S$.

### 4.3 Contours of equal joint exceedance probability with a trivariate copula

We represent in Figure 10 trivariate joint exceedance probability for return periods of 10, 100 and 1.000 years. The trivariate copula used is therefore constructed from a Clayton copula parameter 2.37 connecting $H$ and $T$ and a copula parameter 0.56

connecting $F_{HT}$ and $F_S$.

In order to better visualize the incidence of return periods on trivariate joint exceedance probability, cross-sections along ($H$, $T$), ($H$, $S$) and ($T$, $S$) are shown for $T = T_1$, $H = H_1$ and $S = S_1$ in Figures 10a, 10b and 10c respectively.

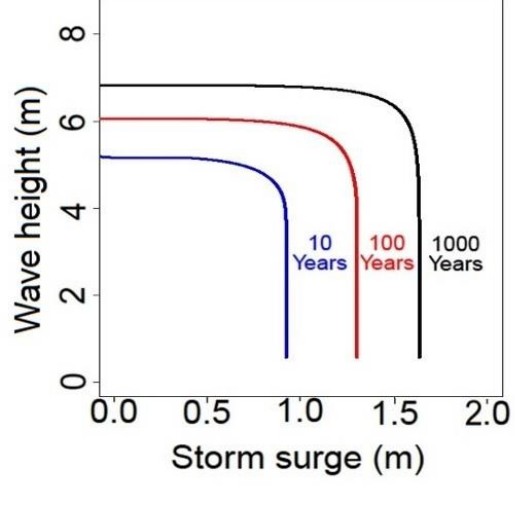

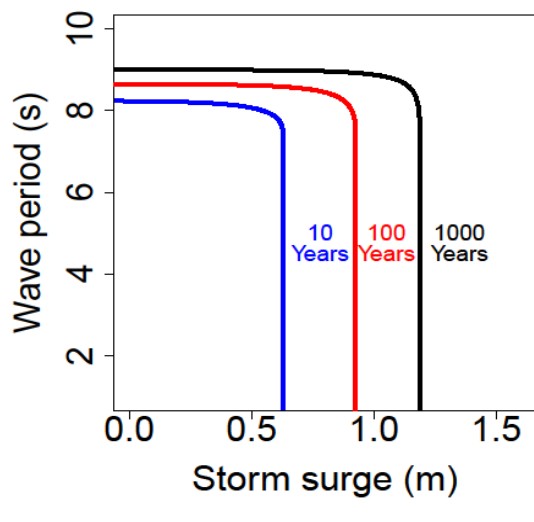

*a)*                    *b)*

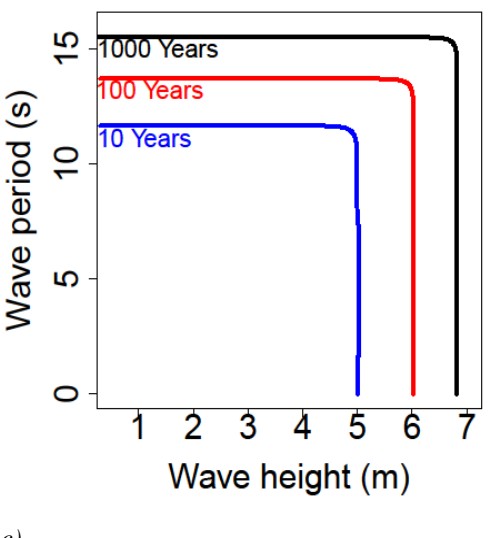
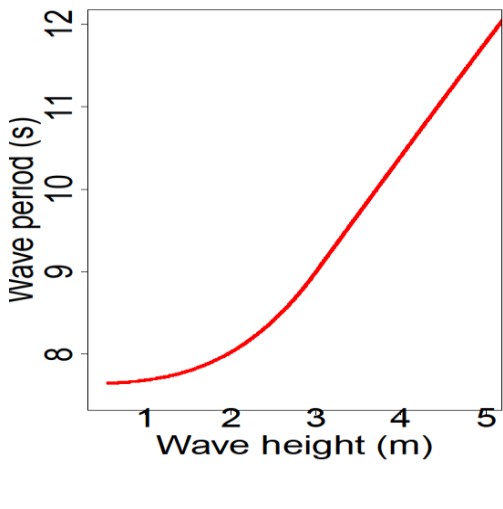

c)                                    d)

**Figure 10** : Contours of equal joint exceedance probability with a trivariate copula.

In Figures 10a, 10b and 10c, a constant wave period, a constant wave height and a constant storm surge respectively are fixed
corresponding to an annual return period. We show the joint exceedance probability of wave height and storm surge, of storm
surge and wave period and of wave height and wave period respectively for three return periods of 10, 100 and 1.000 years.
In the three figures we recognize the usual pattern and the characteristics of a strong correlation for ($H, T$). In Figure 10c we
recognize indeed the classic pattern of contours for very dependent variables. Wave height and wave period are the most
correlated variables. This result is not surprising all the more since we deal with pure wind waves after we have removed the
swell. In Figure 10d, a relationship between $H$ and $T$ is obtained with a trivariate copula with ($H,S$) satisfying a joint exceedance
probability of 1.000 years and with $T$ which maximizes the trivariate joint probability density function. This relationship
enables us to obtain the wave period from the wave height and the storm surge.

**4.4 Error rate and goodness of fit for trivariate copulas**

In order to show the utility of the constructed trivariate copula, we determine the error rate of the different copulas in the Le
Havre area using the formula of the error given by equation (1) and the definition of the error rate given by exp(e) – 1 (see
Table 7).

| Copula | Clayton | Gumbel |
|---|---|---|
| $\overline{C}_2(\overline{C}_1(\overline{F}_H,\overline{F}_s),\overline{F}_T)$ | 6.9 % | |
| $\overline{C}_2(\overline{C}_1(\overline{F}_T,\overline{F}_s),\overline{F}_H)$ | 4.7 % | |
| $\overline{C}_2(\overline{C}_1(\overline{F}_H,\overline{F}_T),\overline{F}_s)$ | **3.8 %** | 22.2 % |
| $\overline{C}(\overline{F}_H,\overline{F}_T,\overline{F}_S)$ | 8.8 % | 169.0 % |

**Table 7** : Error rate of the different trivariate copulas for the port of Le Havre.

The results obtained by the trivariate copula constructed by two bivariate copulas and two parameters are the best of the four
fitted trivariate copulas. However, by aggregating the most correlated variables first, the robustness improves as was stated by
Charpentier (2014).

As expected, with one parameter Archimedean copula is less robust than fully nested hierarchical copula with two parameters.
It can also be seen that by associating the most correlated variables ($H, T$), the Clayton copula gives better results than the
Gumbel copula. For a single parameter the trivariate copula constructed with the Clayton copula is significantly more accurate
than the Gumbel copula.

Table 7 shows finally that the choice of the copula is much more important than the choice of the trivariate method of construction. This result validates our choice of a simple method of construction that can even lead to the most robust results according to Corbella (2013).

| | CHI-2 | KS |
|---|---|---|
| $\overline{C}_2(\overline{C}_1(\overline{F}_H, \overline{F}_T), \overline{F}_S), \theta_1 = 2.37, \theta_2 = 0.56$ | 4.91 | 0.039 |
| $\overline{C}(\overline{C}(\overline{F}_H, \overline{F}_T), \overline{F}_S), \theta = 0.56$ | 5.97 | 0.098 |
| $\overline{C}(\overline{F}_H, \overline{F}_T, \overline{F}_S), \theta = 0.56$ | 5.97 | 0.098 |

**Table 8** : Goodness of fit of the trivariate copulas for the port of Le Havre.

In table 8 is presented the goodness of fit of the trivariate copulas for the port of Le Havre through the chi-squared test (CHI-2) and the Kolmogorv-Smirnov test (KS). The best results are obtained with two parameters. As expected, the fit of the single parameter Archimedean copula and of fully nested hierarchical copula are exactly the same copula as shown in Table 8. The results highlight the contribution of trivariate copulas constructed as a fully nested hierarchical copula with the help of two Clayton bivariate copulas and two parameters by first aggregating the two most correlated parameters.

## 5 Conclusion

Wave structure designers must accurately estimate return periods of parameters such as storm surge, wave height and wave period, and more specifically, their joint probabilities of exceedance. In engineering projects, this joint probability of exceedance is often related to the product of univariate probabilities by means of a simple factor. This method can cause damaging design errors. After highlighting the limitation of the current simplified Defra method, the theory of copula is introduced. Copulas make it possible to couple the marginal laws in order to obtain a multivariate law.

Analysis of the tail dependence of the sample is used to make an initial selection of the copulas. This is because if the sample has lower tail dependence (upper tail dependence, respectively), the copula with the same class of tail dependence or an inverse tail dependence is chosen by taking the survival copula. The correlation between the storm surge and wave height is modelled using the Clayton copula and the survival Gumbel copula.

In order to take into account the three variables (wave height, wave period, and storm surge), we show that a fully nested hierarchical trivariate copula with two parameters is the best construction technique. This function satisfies the mathematical properties of the copulas. The error rate of 3.8 % is lower than the trivariate copula obtained by generalizing the Clayton copula with a single parameter (error rate of 8.8 %). We confirm that the best results are obtained by first aggregating the most correlated variables that are here wave height and wave period. Nevertheless, the choice of method of aggregation is much less important than the choice of the copula.

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

**Appendices**

**Appendix A: Outlines of copula theory**

**A.1 Bivariate cumulative distribution function**

We denote by $F_X$ the cumulative distribution function (CDF) of a random variable defined by

$$F_X(x) = P(X \leq x) = \int_{-\infty}^{x} f_X(y)dy \tag{A.1}$$

where $P$ is the probability.

We also introduce the survival function (SF) denoted by $\bar{F}_X$ and defined by

$$\bar{F}_X(x) = P(X > x) = \int_{x}^{\infty} f_X(y)dy = 1 - F_X(x). \tag{A.2}$$

The survival function is related to the probability density function $f_X$ by

$$f_X(x) = -\frac{d\bar{F}_X(x)}{dx}. \tag{A.3}$$

Our objective is to obtain the bivariate cumulative distribution function $F_{XY}(x,y) = P(X \leq x, Y \leq y)$ or the bivariate survival function $\bar{F}_{XY}(x,y) = P(X > x, Y > y)$. For more information, the reader may refer to (Dodge, 1999; Revuz, 1997; Ouvrard, 1998; Manoukian, 1986).

We must model the correlation between, for example, wave heights $H$ and storm surges $S$ by proposing a relation defining the joint cumulative distribution function from the univariate cumulative distribution functions. We thus seek to obtain a function

$C$ which links the bivariate cumulative distribution frequency $F_{XY}(x,y)$ to the univariate cumulative distribution frequencies $F_X(x)$ and $F_Y(y)$ by integrating a correlation parameter as follows

$$F_{XY}(x,y) = C[F_X(x), F_Y(y)]. \tag{A.4}$$

**A.2 Current practice in coastal engineering**

The simplified Defra method that is presented for example in Ciria *et al.* (2007) makes it possible to directly connect the joint probability density function $f_{XY}$ to the product of the univariate probability density functions $f_X$ and $f_Y$ through a dependence factor denoted FD as follows

$$f_{XY} = \text{FD} f_X f_Y. \tag{A.5}$$

The dependence factor FD depends on the correlation coefficient $\rho$ obtained from the Gaussian copula (see definition in section A.3.2). The variables $X$ and $Y$ for the bivariate analysis are generally wave height $H$ and storm surge $S$. The dependence factor is region specific and results from the analysis of the local correlation between wave heights and storm surges.

The correspondence table between the correlation coefficient $\rho$ and the dependence factor FD is given by Kergadallan (2013). This table recommends, for example, for the North Sea, Channel and Atlantic coast the use of a minimum dependence factor

FD of 25 that is a weak dependence.

**A.3 Copulas**

The copula is a statistical tool to characterize the dependence between several random variables where linear correlations are generally not able to represent them accurately (Charpentier, 2014). According to the latter, copulas have become an important tool for modelling a multivariate law that "couples" univariate cumulative distribution functions, hence the Latin name

"copula" name chosen by Sklar (1959).

If $C$ is the copula associated with a random variable vector $(X, Y)$ then the copula $C$ couples the univariate cumulative distribution functions $F_X(x)$ and $F_Y(y)$ using (A.4).

Survival functions can also be coupled in the sense that there exists a survival copula $\bar{C}$ such that

$$\bar{F}_{XY}(x, y) = \bar{C}[\bar{F}_X(x), \bar{F}_Y(y)]. \tag{A.6}$$

The survival copula $\bar{C}$ is defined from the copula $C$ as follows

$$\bar{C}(\bar{F}_X(x), \bar{F}_Y(y)) = -F_X(x) - F_Y(y) + 1 + C(F_X(x), F_Y(y)). \tag{A.7}$$

In the following description, the univariate cumulative distribution functions $F_X(x)$ and $F_Y(y)$ will be noted $u_1$ and $u_2$ respectively. A copula is a function $C : [0,1]^2 \to [0,1]$ which satisfies the following three conditions

$$
\begin{array}{llll}
i) & C(u_1, 0) = C(0, u_2) = 0 & \forall u_1, u_2 \in [0,1]; \\
ii) & C(u_1, 1) = u_1 \text{ and } C(1, u_2) = u_2 & \forall u_1, u_2 \in [0,1]; & \text{(A.8)} \\
iii) & C(v_1, v_2) + C(u_1, u_2) - C(u_1, v_2) - C(v_1, u_2) \geq 0 & \forall 0 \leq u_i \leq v_i \leq 1.
\end{array}
$$

In the continuation of the paragraph on the description of the copula the functions of distribution $F_X(x)$ and $F_Y(y)$ will be noted $u_1$ and $u_2$.

Sklar (1959) states that there exists a copula $C$ such that for each $x$ and $y$ $F_{XY}(x, y) = C[F_X(x), F_Y(y)]$. If the functions $F_X$ and $F_Y$ are continuous then $C$ is unique. Four of commonly applied copulas families are the Archimedean, Elliptic, Marshall-Olkin and Archimax.

### A.3.1 Archimedean copulas

Archimedean copulas are defined as follows : $\phi$ is a decreasing function convex on $[0,1] \to [0,+\infty[$, as $\phi$ (1) = 0 and $\phi$ (0) = $\infty$. We call a strict Archimedean copula of generator $\phi$ the copula defined as follows

$$C(u_1, u_2) = \phi^{-1}[\phi(u_1) + \phi(u_2)], u_1, u_2 \in [0,1] \tag{A.9}$$

Archimedean copulas have interesting properties, in particular the possibility of aggregating more than two variables as follows

$$C(u_1, u_2, \dots, u_n) = \phi^{-1}[\phi(u_1) + \phi(u_2) + \dots + \phi(u_n)], u_1, u_2, \dots, u_n \in [0,1] \tag{A.10}$$

Archimedean copulas are given in table A1.

| Name | Copula | Generator | Inverse generator |
|---|---|---|---|
| Clayton ($\theta > 0$) | $[u_1^{-\theta} + u_2^{-\theta} - 1]^{-1/\theta}$ | $\dfrac{t^{-\theta} - 1}{\theta}$ | $(1 + \theta t)^{-1/\theta}$ |
| Frank ($\theta \neq 0$) | $-\dfrac{1}{\theta} \ln\left(1 + \dfrac{(e^{-\theta u} - 1)(e^{-\theta v} - 1)}{e^{-\theta} - 1}\right)$ | $-\ln\left(\dfrac{e^{-\theta t} - 1}{e^{-\theta} - 1}\right)$ | $\dfrac{\ln[1 + e^{-t}(e^{-\theta} - 1)]}{\theta}$ |
| Gumbel ($\theta \geq 1$) | $e^{-\left(u_1^\theta + u_2^\theta\right)^{\frac{1}{\theta}}}$ | $(-\ln(t))^\theta$ | $e^{-t^{1/\theta}}$ |
| Independence | $u_1 u_2$ | $-\ln(t)$ | $e^{-t}$ |
| Joe ($\theta \geq 1$) | $1 - [(1 - u_1)^\theta + (1 - u_2)^\theta - (1 - u_1)^\theta (1 - u_2)^\theta]^{\frac{1}{\theta}}$ | $-\ln(1 - (1 - t)^\theta)$ | $1 - (1 - e^{-t})^{1/\theta}$ |
| Ali-Mikhail-Haq ($-1 \leq \theta \leq 1$) | $\dfrac{u_1 u_2}{[1 - \theta(1 - u_1)(1 - u_2)]}$ | $\ln\left(\dfrac{1 - \theta(1 - t)}{t}\right)$ | $\dfrac{1 - \theta}{e^t - \theta}$ |

Table A1: Archimedean copulas

### A.3.2 Elliptic copulas

Elliptic copulas are Gaussian and Student's copulas:

The Gaussian copula is written as follows

$$C(u_1, u_2) = \frac{1}{2\pi\sqrt{1 - \theta^2}} \int_{-\infty}^{\phi^{-1}(u_1)} \int_{-\infty}^{\phi^{-1}(u_2)} \frac{1}{2\pi(1 - \theta^2)^{0.5}} \exp\left(\frac{x^2 - 2\theta xy + y^2}{2(1 - \theta^2)}\right) dxdy, \theta \in [-1, +1] \tag{A.11}$$

with $\phi$ is a distribution function of $X_i$, with $\mathrm{X} = (X_1, X_2, \ldots, X_n)$ a Gaussian random vector ($\mathrm{X} \sim N_v(0, \Sigma)$), where $\Sigma$ is a covariance matrix.

Student's copula is written as follows

$$C(u_1, u_2) = \int_{-\infty}^{t_v^{-1}(u_1)} \int_{-\infty}^{t_v^{-1}(u_2)} \frac{1}{2\pi(1-\theta^2)^{0.5}} \left[1 + \frac{s^2 - 2\theta st + t^2}{2(1-\theta^2)}\right]^{\frac{-(v+2)}{2}} ds \, dt, \theta \in [-1, +1] \tag{A.12}$$

with $t_v$ a distribution function of the univariate Student distribution law with $v$ degrees of freedom.

They are symmetrical copulas. They are widely used in finance. They are implicit and therefore do not have an explicit
analytical form.

### A.3.3 Marshall-Olkin's copula

Marshall-Olkin's copula is written as follows

$$C(u_1, u_2) = \min(u_1{}^a u_2, u_1 u_2{}^b), (a, b) \in [0,1]. \tag{A.13}$$

### A.3.4 Archimax copulas

Archimax copulas include a large number of copulas, including Archimedean copulas.

A bivariate function is an Archimax copula if and only if it is of the form

$$C_{\phi, A}(u_1, u_2) = \phi^{-1} \left[ (\phi(u_1) + \phi(u_2)) A \left( \frac{\phi(u_1)}{\phi(u_1) + \phi(u_2)} \right) \right], \forall \, u_1, u_2 \in [0,1]^2. \tag{A.14}$$

$A : [0,1] \rightarrow [0.5,1]$ such as $\max(t, 1-t) \le A(t) \le 1$ for each $t$ $0 \le t \le 1$.

$\phi : ]0,1[ \rightarrow [0, +\infty[$ is a convex, decreasing function that satisfies $\phi(1) = 0$.

We will adopt the following notation $\phi(0) = \lim_{u \to 0} \phi(t)$ $et$ $\phi^{-1}(s) = 0$, for $s \ge \phi(0)$.

For more information, refer to reference books such as Joe (1997) and Nelsen (1999). The reader may also refer to Clayton (1978).





**Appendix B : Tail dependence of the site**

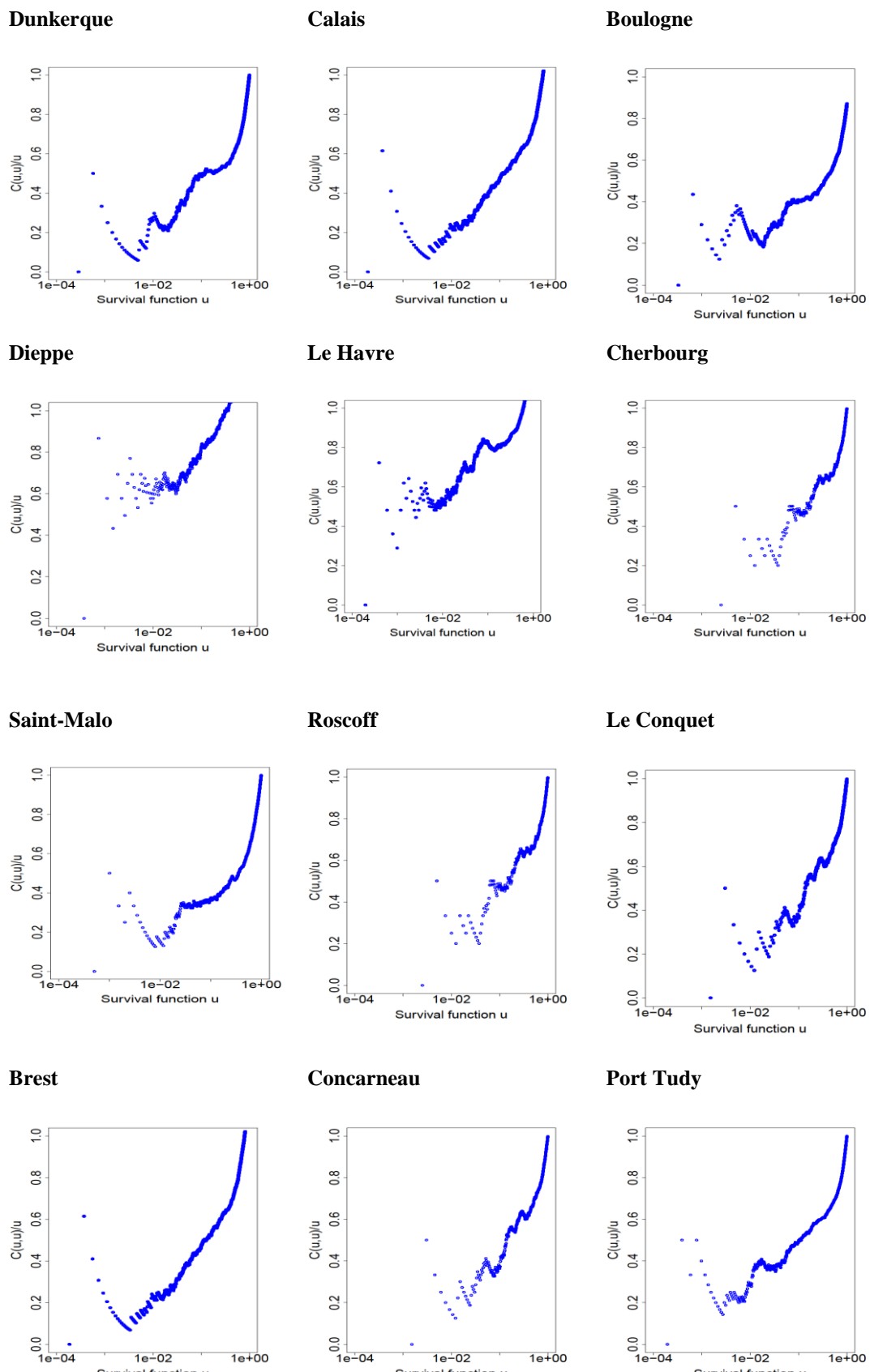

**Saint-Nazaire**  **Saint-Gildas**  **La Rochelle**

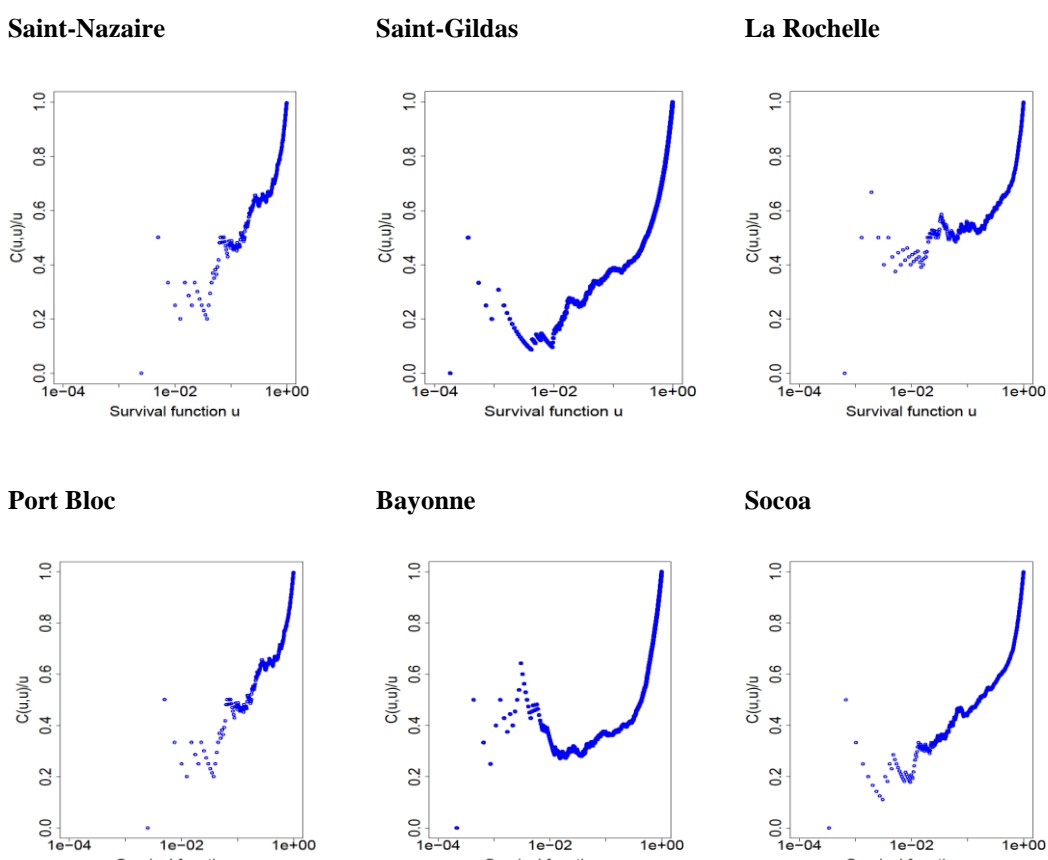

**Port Bloc**  **Bayonne**  **Socoa**

Table B1: Tail dependence of 18 French sites






**Appendix C : Likelihood for 18 French sites**

| Sites | Gumbel | Clayton | Gauss | Franck | Student | Plackett | Joe | AMH | Glambos |
|---|---|---|---|---|---|---|---|---|---|
| **Dunkerque** | 111 | **387** | 244 | 214 | 264 | 226 | 38 | 368 | 125 |
| **Calais** | 90 | **242** | 177 | 172 | 179 | 172 | 23 | 233 | 85 |
| **Boulogne** | 174 | **393** | 287 | 273 | 300 | 279 | 64 | 387 | 164 |
| **Dieppe** | 166 | **383** | 274 | 257 | 286 | 261 | 61 | 379 | 157 |
| **Le Havre** | 352 | **901** | 594 | 551 | 632 | 572 | 117 | 897 | 329 |
| **Cherbourg** | 140 | **383** | 267 | 224 | 277 | 229 | 44 | 317 | 135 |
| **Saint Malo** | 33 | **134** | 79 | 65 | 83 | 67 | 5 | 102 | 32 |
| **Roscoff** | 92 | **273** | 178 | 159 | 188 | 164 | 26 | 229 | 81 |
| **Le Conquet** | 160 | **389** | 28 | 265 | 293 | 268 | 54 | 365 | 150 |
| **Brest** | 178 | **439** | 322 | 295 | 327 | 299 | 59 | 417 | 168 |
| **Concarneau** | 66 | **115** | 97 | 96 | 98 | 94 | 31 | 117 | 64 |
| **Port Tudy** | 391 | 899 | 653 | 627 | 665 | 635 | 139 | **909** | 369 |
| **St Nazaire** | 438 | 1001 | 728 | 713 | 745 | 710 | 159 | **1009** | 522 |
| **Saint Gildas** | 282 | 726 | 492 | 471 | 509 | 479 | 87 | **737** | 265 |
| **La Rochelle** | 107 | **303** | 197 | 186 | 199 | 184 | 30 | **303** | 100 |
| **Bayonne** | 75 | **275** | 153 | 111 | 179 | 116 | 19 | 162 | 67 |
| **Soccoa** | 62 | **230** | 122 | 105 | 155 | 110 | 15 | 163 | 51 |
| **Port Bloc** | 31 | **69** | 47 | 50 | 52 | 53 | 12 | 69 | 28.8 |

755            Table C1: Likelihood for 18 French sites