# Peer review of "Trivariate copula to design coastal structures"

_Natural Hazards and Earth System Sciences, 2020_

## Referee Comment (RC1) · Anonymous Referee #1 · 1 Apr 2020

This paper demonstrates the disparity in the contours of equal joint exceedance probability of wave height and storm surge associated with several bivariate copulas and those obtained using an existing method at two sites in northern France. Two selection criterion are adopted in collaboration with tail dependence coefficients to determine the best fitting among the ten tested copulas. The superiority of (trivariate) nested hierarchical construction over standard trivariate Archimedean copulas for modelling the dependence between wave height, wave period and storm surge is also exemplified at one of the sites.

A significant proportion of the current manuscript is composed of material that can be found elsewhere, while the absence of any discussion on the latest modelling of the joint distribution of the variables comprising an extreme sea state is a glaring omission. Moreover, in parts of the manuscript individual sentences are listed rather than crafted into paragraphs, many figures and tables are poorly explained and there is a lack of referencing throughout. From a technical perspective, although the bivariate results are interesting the trivariate analysis only considers two approaches both of which have been shown to be inferior to pair copula construction for higher dimensional modelling. This reviewer therefore believes that the manuscript will require very significant revisions to be worthy of publication in this special issue of Natural Hazards and Earth System Science.

General comments:

The introduction fails to place the work into the wider context of copula modelling in the field of hydrology or multivariate modeling of extreme sea states carried out to date. The latter discussion should concern work where the dependence between pairs of the wave height H, wave period T and storm surge S or all three are considered (e.g. Gouldby et al. 2014). There is a general lack of referencing throughout the paper.

A coherent and sufficiently detailed explanation of the limitation(s) of the Defra method is also lacking. For instance, does the methods limitations stem from a poor fit of the Gaussian copula from which the dependence factor is derived or the spatial extent covered by each dependence factor a combination of both or other factors.

The word "accuracy" is used repeatedly throughout the paper, however the true shape of the dependence is unknown. Consider replacing "accuracy" with "robustness" or similar. The colloquial [e.g. "variables taken separately" (P1 L31) and "even though this is a complicated exercise" (P2 L42)] and occasionally subjective [e.g. "relatively innovative" (P1 L32)] language used in the manuscript needs remedying.

The "Data Used" subsection feels out of place in the "Theoretical approach" section. Please consider moving the "Data use" subsection to the start of the "Results for bivariate copulas" section. Furthermore, Figure 1 should appear immediately after the first introduction of Le Harve and Saint-Malo in the main body of the manuscript. Perhaps refer to the two sites as two ports in northern France in the introduction so Figure 1 can be placed after the body of text comprising section "2.1 Data Used" in the submitted

manuscript.

The Tables in the results section are often more difficult to interpret than they need to be. To aid interpretation the columns could first be grouped by site i.e., the first half of the columns represent Saint Malo and the second half corresponding to Le Harve.

Sections 2.2 to 2.4 contain material that can be sourced from a multitude of other books/papers. Consider removing or moving to the appendix.

Aas and Berg (2009) show that pair copula construction is less restrictive in terms of the class of copulas that can be mixed and parameter constraints than nested Archimedean construction and are thus more suitable for higher dimensional modeling. The quality of the paper would be elevated substantially if a form of pair copula construction were also fitted in section 4.

The results for trivariate copulas (Section 4) requires more detailed explanation as to the significance of the results. For example, currently Section "4.4 Contours of equal joint exceedance probability with a trivariate copula" is completely devoid of any meaningful discussion of the results.

Often technical concepts or methods e.g., iso-values (P2 L46) or the Chakak and Koehler procedure (P2 L42) are introduced without any or very little introduction.

Specific comments:

P1 L7-8: "The Defra method that is currently used ...". Please detail where the method is currently used.

P1 L9-10: "These schematic correlations do not, however, represent all the complexity of the reality and may lead to damaging errors in coastal structure design." Vague.

P1 L18: Replace "fittest" with "best fitting".

P1 L25-26: "We must therefore address the lack of accuracy of the dependencies between the different variables characterizing the sea state (Sergent et al., 2014; Hawkes,

2005) such as wave height H, wave period T and storm surge S." Please make clear that the "lack of accuracy" refers to the modeling procedure.

P1 26-27: "The design of coastal structures is based in particular on the return periods of wave overtopping or of armour damage.". Reference required.

P1 L35: "Its use in environmental science especially concerns hydrology." Reference required.

P1 L39: "The bivariate return period can be generalized to the multivariate case." Additional explanation or reference required.

P1 L40: "Copulas generally only allow two parameters." Inaccurate.

P2 L46 & P2 L49: "isovalues" or "iso-values". Inconsistent spelling.

P3 L84: "Defra method [2005] ...". Reference not listed in References Section.

P2 L66– P5 L150: I suggest most of this text is move to an appendix.

Table 2: The Student copula does not appear in Table 3 but is mentioned in the text below. P10 L293: "If the sample does not have a tail dependence, then the use of Gaussian copula or Student copula or other copula with the same tail dependence characteristics is recommended." The Student copula possesses tail dependence.

P11 L309 "Until now the simplified Defra method has been quite popular among coastal engineers". Rephrase, too colloquial, also a reference is required. Figure 2: Caption needs more detail. For instance, which site(s) is being considered and which of the methods corresponds to the black line and blue crosses?

Table 3: Typo. "041" in the final row of the table.

Table 3: Caption needs improvement. 'Parameter" column labels needs defining.

Figures 3,4, 5, 6 & 7: Sub-figures need (a) and (b) to explicitly denote correspondence between the plots and the sites.

P12 L350: "The value of the log likelihood of the Gumbel survival copula is large.". Large with respect to what?

P13 L364: "We note Emin the minimum of the error e . . . ". Add "mean" before error.

Table 4: The Emin numbers in the Table do not match the minimum of the mean errors shown in Figure 4. Please check results and, if they should not match the minimums shown in Figure 4 please explain why.

Table 5: Information in Table 5 is recycled from Tables 3 and 4, thus it presents no new information. Remove.

P14 L381: ". . . we show the observed and calculated joint frequencies for the Le Havre sample . . .". Need to add reference to Figure 5(a) here.

P16 L414-415: I believe Figure 6 only contains the results for one rather than both sites.

Figure 6: Adjust Figure to detail the location to which the results refer.

P19 L474-479: Data sources are normally described when the case study site is first introduced.

P19 L480-481: "The copula parameters were calibrated from samples where wave height values less than one meter were excluded, thus reducing the sample size to about 3.000 values". Are the copulas fitted to all pairs/triplets of observations where the wave height exceeded 1 meter? If not, please alter text to clarify.

Figures 8-11: Amalgamate these four Figures into a single Figure.

P20 L490-495 Remove as text already explained in the captions.

References:

Aas, K., and Berg, D.: Models for construction of multivariate dependence – a comparison study, The European Journal of Finance, 15, 7-8, 639-659, 2009.

[Figure]

Gouldby, B., Méndez, F.J., Guanche, Y., Rueda, A. and Mínguez, R., 2014. A methodology for deriving extreme nearshore sea conditions for structural design and flood risk analysis. Coastal Engineering, 88, pp.15-26

---

## Author Comment (AC1) · 30 Apr 2020

Dear editor,

We thank firstly the reviewer #1 for his detailed review and interesting comments. Almost all of his remarks have been taken into account.

Nevertheless, the Aas and Berg (2009) pair copula construction based on conditionnal distribution has not been introduced even it is cited. According to Corbella's (2013) conclusions, valid at least for his application, the use of conditionnal distributions does not improve the results and increases the complexity of treatments.

The originality of the paper relies on the use of a fully nested hierarchical copula with two parameters, on the analysis of the order of aggregation of random variables and the simultaneous sensitivity to the choice of the copula and to the construction of the

trivariate copula.

**Trivariate copula to design coastal structures**

Olivier Orcel[1], Philippe Sergent[1], François Ropert[1]

[1]Cerema, Margny-Lès-Compiègne, 60280, France

*Correspondence to*: Philippe Sergent (philippe.sergent@cerema.fr)

[revised manuscript text omitted]

---

## Referee Comment (RC2) · Anonymous Referee #2 · 5 May 2020

First, I wish to apologize for the delay in transmitting the review, but a lockdown at home with a baby around is not the most favourable environment for such a work... Having taking note of the first review, written by somebody that is much more expert than I am in the theory (and application) of multivariate extremes, I will simply complement his review with additional comments.

This paper aims at replacing the so-called Defra method currently practiced in northern France by an approach relying upon the use of bivariate and trivariate copulas. Some statistical tools that are presented are quite useful and could indeed improve the current practice. However, in addition to the comments posted by the first reviewer, I believe that some significant improvements are requested as regards sampling and event definition, the choice of the copulas, the analysis of tail dependence and the signification of the return period.

[Figure]

First, the so-called Defra method should not be presented as "state of the art", in particular for the simplified version proposed by Kergadallan with the dependence factor. If this is "current practice", it should be specified "where" and "by who".

A crucial point is the sampling, and hence the event definition. The choice of the values at high tide certainly has its justification if the final purpose is wave overtopping or coastal flooding. However, this is not the only one. It does not consider extreme sea states or surges occurring around low tide, even though it may be valuable information. For instance, Kergadallan (2015) recommends selecting the maximum Hs value within a time window centred on the time of high water. Furthermore, it yields quite a large sample (706 events per year) and low to moderate values may be overweighed in the sample. A threshold on Hs may be applied to reduce sample size. Last, the sample should be made of independent and identically distributed (i.i.d.) tuples. Is the independence assumption valid when two tuples per day are selected? Is there only one wave population, or in other words is the extreme behaviour of waves similar for storms from the west or from the north-east? The topic of event definition in such a context (waves / level in coastal areas) is discussed by Hawkes (2002) and Mazas (2017, 2019), among others.

As regards tail dependence, the authors rightly present both the lower and upper tail dependences, and the fact that copulas with the same structure of dependence as the sample of observations. But surprisingly, they focus on the lower tail dependence only for the choice of the copula. Because they find a (weak) lower tail dependence, they choose copulas that will fit best... the least interesting part of the sample! Why not assessing the upper tail dependence, and possibly include extreme value copulas (a special case of Archimax copulas) such as Gumbel-Hougaard, Galambos or Hüsler-Reiss copulas? See for instance Mazas and Hamm (2017) for an application of these copulas for Hs / surge modelling.

Another concern is the return period, a topic intimately lonked to sampling / event definition. First, the return period of "source phenomena" such as Hs / sea level is a very

different thing than the return period of "response phenomena", as discussed among many others by Hawkes et al. (2002) or Mazas (2019). Therefore, when writing in the introduction (l. 26-27) that "the design of coastal structures is based in particular on the return periods of wave overtopping or of armour damage", the authors should acknowledge that they do not address the return period of such phenomena in the paper. Second, there are several definitions of return period (that is a yearly probability of exceedance) in the bivariate case, let alone the trivariate one: see in particular Serinaldi (2015) and Haselsteiner et al. (2017) who detail the different types of environmental contours with respect to the definition of the return period (i.e. the definition of the bivariate probability to consider). In this paper, the authors consider the joint exceedance probability and the associated contours, which is of course quite a relevant choice; however, it should be recalled that this is not the only one possible.

I believe that, once all these points (along with the ones presented by the other reviewer) are clarified, the interest of a trivariate modelling will appear much more plainly.

Specific comments:

- l. 43, "incompatibility problem": maybe a very short explanation of what it means would help

- l. 56: to be accurate, the random variables are "Hs (resp. T, S) at high tide" (see discussion on sampling and event definition).

- l. 63-65: a short description of the mixture model would be welcome

- section 2.3: explain in which context the Defra method is "current practice"

- l. 92: please specify that FD=25 corresponds to "weak dependence"

- section 3.1: change the title of the section

- l. 312-313: the value of FD=20 is lower than the minimal value of FD=25 recommended by Kergadallan

- Figure 6 really needs some improvement, I have not understood it

References

Haselsteiner, A. F., Ohlendorf, J. H., Wosniok, W., & Thoben, K. D. (2017). Deriving environmental contours from highest density regions. Coastal Engineering, 123, 42-51.

Hawkes, P.J., Gouldby, B.P., Tawn, J.A., Owen, M.W., 2002. The joint probability of waves and water levels in coastal engineering design. Journal of Hydraulic Research 40, 241–251.

Hawkes, P., 2008. Joint probability analysis for estimation of extremes. Journal of Hydraulic Research 46, 246–256.

Kergadallan X., Estimation Des Niveaux Marins Extrêmes Avec Et Sans L'action Des Vagues Le Long Du Littoral Métropolitain (Doctoral dissertation), Université Paris-Est, 2015.

Mazas, F., 2019. Extreme events: a framework for assessing natural hazards. Nat Hazards 98, 823–848.

Mazas, F., Hamm, L., 2017. An event-based approach for extreme joint probabilities of waves and sea levels. Coastal Engineering 122, 44–59. https://doi.org/10.1016/j.coastaleng.2017.02.003

Serinaldi, F., 2015. Dismissing return periods! Stochastic Environmental Research and Risk Assessment 29, 1179–1189. https://doi.org/10.1007/s00477-014-0916-1

---

## Author Comment (AC2) · 16 Jun 2020

According to the comments of the referee, we have modified the paper as follows.

The Defra method is now presented as a current practice. The section 3.2 explains in which context the Defra method is a current practice. We have recalled that FD=25 is a weak dependence and the FD=20 is lower than the value that is recommended by Kergadallan. In section 2.1, we acknowledge that the choice of the values at high tide is not the only choice. We had omitted to mention that we used the same data as Kergadallan and his own method as we selected the maximum Hs value within a time window centred on the time of high water. That is now mentioned in the paper. We nevertheless consider that a threshold on Hs is inappropriate in regards of the distribution function (this threshold is applied for copula but not for the distribution function). Since we have two wave populations, we have indeed used a threshold and excluded

[Figure]

wave height values less than one meter (see figure 2). We acknowledge that the independence assumption is not completely valid when two tuples per day are selected but this is a common assumption. Full compliance with independence would lead to ignore some relevant pairs of wave height and surge values. We focus on the lower tail dependence of the survival copula. That is now better explained in section 3.3. We choose the survival copula instead of the standard copula because it simplifies the equations (22), (26), (30), (34). We acknowledge that we do not address the return period of wave overtopping and of armour damage. We also recall that the definition of the return period is not unique. The mixture model is similar to Chakak and Koehler (1995) method that is explained line 50. Its compatibility problem is explained line 51. The title of section 3.1 is changed. The caption of figure 7 is completed in order to improve the understanding of the figure. Five proposed references are added to the text.

---

## Author Comment (AC3) · 16 Jun 2020

Please find herewith : - the revised manuscript ; - the reply to referees.

**Trivariate copula to design coastal structures**

Olivier Orcel[1], Philippe Sergent[1], François Ropert[1]

[1]Cerema, Margny-Lès-Compiègne, 60280, France

*Correspondence to*: Philippe Sergent (philippe.sergent@cerema.fr)

5 **Abstract.** Some coastal structures must be redesigned in the future due to rising sea levels caused by global warming. The design of structures subjected to the actions of waves requires an accurate estimate of the long return period of such parameters as wave height, wave period, storm surge and more specifically their joint exceedance probabilities. The simplified Defra method that is currently used in particular for European coastal structures makes it possible to directly connect the joint exceedance probabilities to the product of the univariate probabilities by means of a single factor. These schematic correlations
10 do not, however, represent all the complexity of the reality because of the use of this single factor. That may lead to damaging errors in coastal structure design. The aim of this paper is therefore to remedy the lack of robustness of these current approaches. To this end, we use copula theory with a copula function that aggregates joint distribution function to its univariate margins. We select a bivariate copula that is adapted to our application by the likelihood method with a copula parameter that is obtained by the error method. In order to integrate extreme events, we also resort to the notion of tail dependence. We select
15 the copulas with the same tail dependence as data. In the event of an opposite tail dependence structure, we resort to the survival copula. The tail dependence parameter makes it possible to estimate the optimal copula parameter. The most robust copulas for our practical case with applications in Saint-Malo and Le Havre (in Northern France) are the Clayton normal copula and the Gumbel survival copula. The originality of this paper is the creation of a new and robust trivariate copula with an analysis of the sensitivity to the method of construction and to the choice of the copula. Firstly, we select the best fitting of the
20 bivariate copula with its parameter for the two most correlated univariate margins. Secondly, we build a trivariate function. For this purpose, we aggregate the bivariate function with the remaining univariate margin with its parameter. We show that this trivariate function satisfies the mathematical properties of the copula. We finally represent joint trivariate exceedance probabilities for a return period of 10, 100 and 1000 years. We finally conclude that the choice of the bivariate copula is more important for the accuracy of the trivariate copula than its own construction.

25 **1 Introduction**

The design of coastal structures requires the multiplicity of variables and their degree of correlation to be taken into account. We must therefore address the lack of robustness in the modelling procedure of the dependencies between the different variables characterizing the sea state (Sergent *et al.*, 2014; Hawkes, 2005) such as wave height $H$, wave period $T$ and storm surge $S$. The design of coastal structures is based in particular on the return periods of wave overtopping or of armour damage
30 (Ciria *et al.*, 2007). Since the applications on wave overtopping and armour damage depend on the parameters of the coastal structure, we will not deal with the return periods of these quantities. The aim of this paper is however to improve the methods of estimating them in order to avoid costly and inappropriate decisions (Li *et al.*, 2008). To this end, we provide accurate estimates of the correlations between the variables $H$, $T$ and $S$ and obtain reliable return periods. Currently, in reference manuals such as the Rock Manual (Ciria *et al.*, 2007), it is recommended that a factor be applied to the product of univariate survival
35 functions in order to determine the joint period. Copulas are mathematical tools for modelling the dependence structure of several random variables. The theory of copulas was developed by the mathematician Sklar (1959). The copula is a written form of the joint distribution function that provides all the information on the dependency structure. The recent interest in copulas started in financial risk management and insurance. Its use in environmental science especially concerns hydrology

**Fig. 1.**
**Reply to Referee #1**

Several references are added in the paper. Colloquial language is removed. Sections 2.2 to 2.4 are moved to the appendix. Figure 1 appears now immediately after the first introduction of Le Havre and Saint-Malo in the main body of the manuscript. The columns of the tables are grouped by site. More detailed explanations of the results are added.

The limitations of the Defra method is commented is section 3.2. They are due to the use of the simplified version of the Defra method but also to the choice of a Gauss copula without tail dependence.

We mention now Aas and Berg (2009) who propose copula construction with conditional sets : the pair copula construction (PCC). As the bivariate copulas that are selected as the most promising in our application are Archimedean copulas, simpler methods of construction are available. We find that it is useless to use more complex techniques with compatibility problem and that have been less robust than fully nested method in some applications.

**Reply to Referee #2**

The Defra method is now presented as a current practice. The section 3.2 explains in which context the Defra method is a current practice. We have recalled that FD=25 is a weak dependence and the FD=20 is lower than the value that is recommended by Kergadallan.

In section 2.1, we acknowledge that the choice of the values at high tide is not the only choice. We had omitted to mention that we used the same data as Kergadallan and his own method as we selected the maximum Hs value within a time window centred on the time of high water. That is now mentioned in the paper.

We nevertheless consider that a threshold on Hs is inappropriate in regards of the distribution function (this threshold is applied for copula but not for the distribution function). Since we have two wave populations, we have indeed used a threshold and excluded wave height values less than one meter (see figure 2). We acknowledge that the independence assumption is not completely valid when two tuples per day are selected but this is a common assumption. Full compliance with independence would lead to ignore some relevant pairs of wave height and surge values.

We focus on the lower tail dependence of the survival copula. That is now better explained in section 3.3. We choose the survival copula instead of the standard copula because it simplifies the equations (22), (26), (30), (34).

We acknowledge that we do not address the return period of wave overtopping and of armour damage. We also recall that the definition of the return period is not unique.

The mixture model is similar to Chakak and Koehler (1995) method that is explained line 50. Its compatibility problem is explained line 51. The title of section 3.1 is changed. The caption of figure 7 is completed in order to improve the understanding of the figure. Five proposed references are added to the text.

**Fig. 2.**

---

## Author Response (AR1)

**Reply to Referee #1**

A significant proportion of the current manuscript is composed of material that can be found elsewhere, while the absence of any discussion on the latest modelling of the joint distribution of the variables comprising an extreme sea state is a glaring omission. Moreover, in parts of the manuscript individual sentences are listed rather than crafted into paragraphs, many figures and tables are poorly explained and there is a lack of referencing throughout.

*Sections 2.2 to 2.4 are moved to the appendix P22-24 L570 to L653. Several references on the latest modelling of the joint distribution are added in the paper in introduction. We improve the explanations of figures by writing a) Saint-Malo b) Le Havre P11 Figure 4 P12 Figure 5 P15 Figure7, Figure8 and P18 Figure 9. The columns of the tables are therefore grouped by site. More detailed explanations of the results are added. We improve the explanation of Table2 P11. We add 14 references P20-21 L498, 500, 504, 513, 519, 523, 525, 527, 533, 537, 542, 544, 552, 561.*

From a technical perspective, although the bivariate results are interesting the trivariate analysis only considers two approaches both of which have been shown to be inferior to pair copula construction for higher dimensional modelling.

*As the bivariate copulas that are selected as the most promising in our application are Archimedean copulas, simpler methods of construction are available. We find that it is useless to use more complex techniques like Pair-Copula Constructions (PCC) with compatibility problem and that have been less robust than fully nested method in some applications.*

The introduction fails to place the work into the wider context of copula modelling in the field of hydrology or multivariate modeling of extreme sea states carried out to date. The latter discussion should concern work where the dependence between pairs of the wave height H, wave period T and storm surge S or all three are considered (e.g. Gouldby et al. 2014). There is a general lack of referencing throughout the paper.

*Several references on the latest modelling of the joint distribution are added in the paper in introduction.*

A coherent and sufficiently detailed explanation of the limitation(s) of the Defra method is also lacking. For instance, does the methods limitations stem from a poor fit of the Gaussian copula from which the dependence factor is derived or the spatial extent covered by each dependence factor a combination of both or other factors.

*The limitations of the Defra method is commented is section 3.2. They are due to the use of the simplified version of the Defra method but also to the choice of a Gauss copula without tail dependence.*

The word "accuracy" is used repeatedly throughout the paper, however the true shape of the dependence is unknown. Consider replacing "accuracy" with "robustness" or similar. The colloquial [e.g. "variables taken separately" (P1 L31) and "even though this is a complicated exercise" (P2 L42)] and occasionally subjective [e.g. "relatively innovative" (P1 L32)] language used in the manuscript needs remedying.

*Colloquial language is removed and in particular the word "accuracy".*

The "Data Used" subsection feels out of place in the "Theoretical approach" section. Please consider moving the "Data use" subsection to the start of the "Results for bivariate copulas" section. Furthermore, Figure 1 should appear immediately after the first introduction of Le Havre and Saint-Malo in the main body of the manuscript. Perhaps refer to the two sites as two ports in northern France in the introduction so Figure 1 can be placed after the body of text comprising section "2.1 Data Used" in the submitted manuscript.

*The description of data is now in subsection 2.1 named "sets of data". Figure 1 appears now immediately after the first introduction of Le Havre and Saint-Malo in the main body of the manuscript.*

The Tables in the results section are often more difficult to interpret than they need to be. To aid interpretation the columns could first be grouped by site i.e., the first half of the columns represent Saint-Malo and the second half corresponding to Le Havre. Sections 2.2 to 2.4 contain material that can be sourced from a multitude of other books/papers. Consider removing or moving to the appendix.

*The columns of the tables are grouped by site. Sections 2.2 to 2.4 are moved to the appendix.*

Aas and Berg (2009) show that pair copula construction is less restrictive in terms of the class of copulas that can be mixed and parameter constraints than nested Archimedean construction and are thus more suitable for higher dimensional modeling. The quality of the paper would be elevated substantially if a form of pair copula construction were also fitted in section 4.

*We mention now Aas and Berg (2009) who propose copula construction with conditional sets : the Pair-Copula Constructions (PCC). As the bivariate copulas that are selected as the most promising in our application are Archimedean copulas, simpler methods of construction are available. We find that it is useless to use more complex techniques with compatibility problem and that have been less robust than fully nested method in some applications.*

The results for trivariate copulas (Section 4) requires more detailed explanation as to the significance of the results. For example, currently Section "4.4 Contours of equal joint exceedance probability with a trivariate copula" is completely devoid of any meaningful discussion of the results.

*We mention in present Section 3.3 that we recognize the characteristics of a strong correlation for (H, T) in contours of equal joint exceedance. The three main conclusions of the section are as follows:*

- *By aggregating the most correlated variables first, the robustness improves.*
- *As expected, with one parameter Archimedean copula is less robust than fully nested hierarchical copula with two parameters.*
- *Table 7 shows finally that the choice of the copula is much more important than the choice of the trivariate method of construction.*

Often technical concepts or methods e.g., iso-values (P2 L46) or the Chakak and Koehler procedure (P2 L42) are introduced without any or very little introduction.

*We explain that the Chakak and Koehler (1995) method is based on bivariate conditional distributions.*

*We write P2 L42 :*

*''In the literature the Chakak and Koehler (1995) method is commonly used and in particular by Joe (1997) and Salvadori et al. (2007). This method is based on bivariate conditional distributions and requires the use of three bivariate copulas. The method has a compatibility problem. There is no guarantee that the method gives the same result when the order of variables is changed. Aas and Berg (2009) propose copula construction with conditional sets : the pair copula construction (PCC). As the bivariate copulas that are selected as the most promising in our application are Archimedean copulas, simpler methods of construction are available.''*

P1 L7-8 : ''The Defra method that is currently used . . .''  Please detail where this method is currently used.

*We write P1 L7-L8 : The simplified Defra method that is currently used in particular for* *European coastal structures* *makes it possible to directly connect the joint exceedance probabilities to the product of the univariate probabilities by means of a single factor.*

P1 L9-10: "These schematic correlations do not, however, represent all the complexity of the reality and may lead to damaging errors in coastal structure design." Vague.

*We write P1 L9-L10 : These schematic correlations do not however represent all the complexity of the reality* *because of the use of this single factor.**''*

P1 L18: Replace "fittest" with "best fitting".

*We write P1 L19 : best fitting.*

P1 L25-26: "We must therefore address the lack of accuracy of the dependencies between the different variables characterizing the sea state (Sergent et al., 2014; Hawkes, 2005) such as wave height H, wave period T and storm surge S." Please make clear that the "lack of accuracy" refers to the modeling procedure.

*We write P1 L27 : ''We must therefore address the lack of robustness in the modelling procedure of the dependencies between the different variables characterizing the sea state (Sergent et al., 2014; Hawkes, 2005) such as wave height H, wave period T and storm surge S.''*

P1 26-27: "The design of coastal structures is based in particular on the return periods of wave overtopping or of armour damage.". Reference required.

*We write P1 L29 : ''The design of coastal structures is based in particular on the return periods of wave overtopping or of armour damage (Ciria et al., 2007).''*

P1 L35: "Its use in environmental science especially concerns hydrology." Reference required.

*We write P1 L38, 39 : ''Its use in environmental science especially concerns hydrology with the works for example of De Michele and Salvadori (2003), Favre et al. (2004), Grimaldi and Serinaldi (2006), Genest and Favre (2007), Zhang and Singh (2007), Aghakouchak et al. (2010), Lee et al. (2013), Chang et al. (2016).*

P1 L39: "The bivariate return period can be generalized to the multivariate case." Additional explanation or reference required.

*We write P1 L44 : ''The bivariate return period can be generalised to the multivariate case (Charpentier, 2014)''.*

P1 L40: "Copulas generally only allow two parameters." Inaccurate.

*We write P1 L46 : ''Copulas aggregate only two random variables''.*

P2 L46 & P2 L49: "isovalues" or "iso-values". Inconsistent spelling.

*We write P2 L66 L69 : ''Isovalue lines''.*

P3 L84: "Defra method [2005] : : :". Reference not listed in References Section.

*We write : "The use of the simplified Defra method in Ciria et al. (2007)''. We can find the reference P22 L507 Ciria et al.*

P2 L66– P5 L150: I suggest most of this text is move to an appendix.

*We move this text to an appendix. : P22 L568 – P24 L653.*

Table 2: The Student copula does not appear in Table 3 but is mentioned in the text below. P10 L293: "If the sample does not have a tail dependence, then the use of Gaussian copula or Student copula or other copula with the same tail dependence characteristics is recommended." The Student copula possesses tail dependence.

*The Student copula is removed P8 L244.*

P11 L309 "Until now the simplified Defra method has been quite popular among coastal engineers". Rephrase, too colloquial, also a reference is required.

*Section 3.2 is rewritten. The reference Rock Manual (Ciria et al., 2007) is added.*

Figure 2: Caption needs more detail. For instance, which site(s) is being considered and which of the methods corresponds to the black line and blue crosses?

*We add in the caption P10 L284 : for the Havre. The blue crosses correspond to the Defra method as Figure and black line to Fcal = Fmes : the exact value.*

Table 3: Typo. "041" in the final row of the table.

*We write P11 Table 2 : 41.*

Table 3: Caption needs improvement. 'Parameter' column labels needs defining.

*We write P11 Table 2 copula parameter as column label.*

Figures 3, 4, 5, 6 & 7: Sub-figures need (a) and (b) to explicitly denote correspondence between the plots and the sites.

*We write (a) and (b) to explicitly denote correspondance between the plot and the sites.*

P12 L350: "The value of the log likelihood of the Gumbel survival copula is large.". Large with respect to what?

*We write P12 L323 : ''The value of the log-likelihood of the Gumbel survival copula is as large as the log-likelihood of the Clayton copula''.*

P13 L364: "We note Emin the minimum of the error e : : : ". Add "mean" before error.

*We write P13 L339 : ''We note Emin the minimum of the mean error''.*

Table 4: The Emin numbers in the Table do not match the minimum of the mean errors shown in Figure 4. Please check results and, if they should not match the minimums shown in Figure 4 please explain why.

*We modify Figure 5 P12 and Table 3 P13. The numbers in Figure 5 and Table 3 match now.*

Table 5: Information in Table 5 is recycled from Tables 3 and 4, thus it presents no new information. Remove.

*The table is removed.*

P14 L381: ": : : we show the observed and calculated joint frequencies for the Le Havre sample : : :". Need to add reference to Figure 5(a) here.

*We write : "we show in Figure 6 the observed and calculated joint frequencies".*

P16 L414-415: I believe Figure 6 only contains the results for one rather than both sites. Figure 6: Adjust Figure to detail the location to which the results refer.

*It is exact. We modify and write before Figure 7 P14 L 382 Le Havre (3040 values). We add Le Havre in the caption of Figure 7 P15 L384.*

P19 L474-479: Data sources are normally described when the case study site is first introduced.

*We move this text to P9 L256-266 and add P9 Figure 2 – Set of wave data in Le Havre (1979-2002).*

P19 L480-481: "The copula parameters were calibrated from samples where wave height values less than one meter were excluded, thus reducing the sample size to about 3.000 values". Are the copulas fitted to all pairs/triplets of observations where the wave height exceeded 1 meter? If not, please alter text to clarify.

*It is exact. We add P9 L265 : ''The copulas are fitted to all pairs/triplets of observations where the wave height exceeds one meter.'' We add P9 Figure 2 – Set of wave data in Le Havre (1979-2002) in order to show the set of data that is excluded.*

Figures 8-11: Amalgamate these four Figures into a single Figure.

*We almagate these four figures into a single Figure : p14 Figure 6.*

P20 L490-495 Remove as text already explained in the captions.

*We suppress the comment of each figure and write one comment P18 : ''Contours of equal joint exceedance probability with a trivariate copula''.*

Aas, K., and Berg, D.: Models for construction of multivariate dependence – a comparison study, The European Journal of Finance, 15, 7-8, 639-659, 2009.

*We add this reference p20 L578.*

Gouldby, B., Méndez, F.J., Guanche, Y., Rueda, A. and Mínguez, R., 2014. A methodology for deriving extreme nearshore sea conditions for structural design and flood risk analysis. Coastal Engineering, 88, pp.15-26

*We add this reference p21 L525.*

**Reply to Referee #2**

First, the so-called Defra method should not be presented as "state of the art", in particular for the simplified version proposed by Kergadallan with the dependence factor. If this is "current practice", it should be specified "where" and "by who".

*The Defra method is now presented as a current practice. The section 3.2 explains in which context the Defra method is a current practice. P9 L268 is the section 3.2 that describes the current practice. We write : "The use of the simplified Defra method in Ciria et al. (2007) is common among European coastal engineers for the study of wave overtopping or armor damages in coastal structures.".*

A crucial point is the sampling, and hence the event definition. The choice of the values at high tide certainly has its justification if the final purpose is wave overtopping or coastal flooding. However, this is not the only one. It does not consider extreme sea states or surges occurring around low tide, even though it may be valuable information.

*In section 2.1, we acknowledge that the choice of the values at high tide is not the only choice.*

For instance, Kergadallan (2015) recommends selecting the maximum Hs value within a time window centred on the time of high water.

*We had omitted to mention that we used the same data as Kergadallan and his own method as we selected the maximum Hs value within a time window centred on the time of high water. That is now mentioned in the paper. We write P3 L101 : "Kergadallan (2015) recommends selecting the maximum H value within a time window centered on the time of high water. Using the same data, this recommendation is followed.".*

Furthermore, it yields quite a large sample (706 events per year) and low to moderate values may be overweighed in the sample. A threshold on Hs may be applied to reduce sample size.

*We nevertheless consider that a threshold on Hs is inappropriate in regards of the distribution function (this threshold is applied for copula but not for the distribution function).*

Last, the sample should be made of independent and identically distributed (i.i.d.) tuples. Is the independence assumption valid when two tuples per day are selected?

*We acknowledge that the independence assumption is not completely valid when two tuples per day are selected but this is a common assumption. Full compliance with independence would lead to ignore some relevant pairs of wave height and surge values.*

Is there only one wave population, or in other words is the extreme behaviour of waves similar for storms from the west or from the north-east? The topic of event definition in such a context (waves / level in coastal areas) is discussed by Hawkes (2002) and Mazas (2017, 2019), among others.

*Since we have two wave populations, we have indeed used a threshold and excluded wave height values less than one meter (see P9 Figure 2). The references of discussions by Hawkes (2002) and Mazas (2017, 2019) are added.*

As regards tail dependence, the authors rightly present both the lower and upper taildependences, and the fact that copulas with the same structure of dependence as the sample of observations. But surprisingly, they focus on the lower tail dependence only for the choice of the copula. Because they find a (weak) lower tail dependence, they choose copulas that will fit best: : : the least interesting part of the sample! Why not assessing the upper tail dependence, and possibly include extreme value copulas (a special case of Archimax copulas) such as Gumbel-Hougaard, Galambos or Hüsler- Reiss copulas? See for instance Mazas and Hamm (2017) for an application of these copulas for Hs / surge modelling.

*We focus on the lower tail dependence of the survival copula. That is now better explained in section 3.3. We choose the survival copula instead of the standard copula because it simplifies the equations (22), (26), (30), (34).*

Another concern is the return period, a topic intimately lonked to sampling / event definition. First, the return period of "source phenomena" such as Hs / sea level is a very different thing than the return period of "response phenomena", as discussed among many others by Hawkes et al. (2002) or Mazas (2019). Therefore, when writing in the introduction (l. 26-27) that "the design of coastal structures is based in particular on the return periods of wave overtopping or of armour damage", the authors should acknowledge that they do not address the return period of such phenomena in the paper.

*We acknowledge that we do not address the return period of wave overtopping and of armour damage. We write P1 L30: "Since the applications on wave overtopping and armour damage depend on the parameters of the coastal structure, we will not deal with the return periods of these quantities.".*

Second, there are several definitions of return period (that is a yearly probability of exceedance) in the bivariate case, let alone the trivariate one: see in particular Serinaldi (2015) and Haselsteiner et al. (2017) who detail the different types of environmental contours with respect to the definition of the return period (i.e. the definition of the bivariate probability to consider). In this paper, the authors consider the joint exceedance probability and the associated contours, which is of course quite a relevant choice; however, it should be recalled that this is not the only one possible.

*We also recall that the definition of the return period is not unique. We write P2 L76 :" As mentioned by Serinaldi (2015), this option is not unique and will lead to a specific return period that he denotes TAND.".*

l. 43, "incompatibility problem": maybe a very short explanation of what it means would help

*The mixture model is similar to Chakak and Koehler (1995) method that is explained P2 L50. Its compatibility problem is explained P2 L51.*

l. 56: to be accurate, the random variables are "Hs (resp. T, S) at high tide" (see discussion on sampling and event definition).

*We add some details on sampling and event definition (see above).*

l. 63-65: a short description of the mixture model would be welcome

*P2 L50 we write "This method is based on bivariate conditional distributions and requires the use of three bivariate copulas. The method has a compatibility problem. There is no guarantee that the method gives the same result when the order of variables is changed. "*

Section 2.3: explain in which context the Defra method is "current practice"

*The Defra method is now presented as a current practice. The section 3.2 explains in which context the Defra method is a current practice. P9 L268 is the section 3.2 that describes the current practice. We write : "The use of the simplified Defra method in Ciria et al. (2007) is common among European coastal engineers for the study of wave overtopping or armor damages in coastal structures.".*

L92: please specify that FD=25 corresponds to "weak dependence". l. 312-313: the value of FD=20 is lower than the minimal value of FD=25 recommended by Kergadallan

*We recall that FD=25 is a weak dependence and the FD=20 is lower than the value that is recommended by Kergadallan.*

*See P9 L274 "In France, the order of magnitude for the FD coefficient is about 20. Kergadallan (2013) recommends however a minimum value of 25.".*

*See P22 L596 "This table recommends, for example, for the North Sea, Channel and Atlantic coast the use of a minimum dependence factor FD of 25 that is a weak dependence.".*

Section 3.1: change the title of the section

*The title of section 3.1 now 3.2 is changed as "Current pratice : Defra method"*

Figure 6 really needs some improvement, I have not understood it

*The caption of Figure 7 is completed in order to improve the understanding of the figure.*

References

*Five proposed references are added to the text.*

[revised manuscript text omitted]

---

## Referee Report (RR1)

**Review of revised manuscript nhess-2020-80**

The authors have fairly improved their manuscript with a better structure, more references and appreciable rephrasing. The methodology used for obtaining a trivariate distribution is clearer, and the result could be quite useful for coastal engineers in particular when wave period plays a role in addition to wave height and sea level.

However, I think that some points could be better explained, while I remain quite disturbed by one choice of the authors.

As regards the possible clarifications:

- Section 2, l.76: the return period T_AND is introduced, but not defined. The associated probability of joint exceedance should be mentioned. Similarly, this probability could be explicated in section 2.4, l 177.
- the description of the constitution of the sample from the time series is confused, in particular l. 88 to 102. First, present the time series (besides, you mention the measurement network CANDHIS for wave data, while later you use a series from the numerical database Anemoc). Then, explain and justify how and why you build the event sample of high tide values, including the recommendation by Kergadallan. Last, describe the modelling of the marginal distributions (empirical + exponential).
- section 4.2: for the construction of the trivariate copula, make clearer that you use the fully nested hierarchical copula method, and not the first approach discussed in section 2.3.2
- section 2.5: the method you propose for assessing the sample dependence refers only to lower tail dependence. Furthermore, other methods exist such as the chi-plot proposed by Fisher and Switzer (1985, 2001), used in coastal analyses by Mazas (2017) for instance.

My biggest concern is related to the assessment of the sample dependence, and its consequences on the choice of the copula. You assess the lower tail dependence of your two samples, and find a moderate one. This is enough for you to justify the use of Clayton and survival Gumbel copulas. But you do not show that the samples have no (or negligible, or even smaller) upper tail dependence! because you are interested in extreme values (large H, T and S), I still think you focus on the wrong tail. At the very least, you should justify that lower tail dependence is more important than upper tail dependence for your analysis.

Last, I think that physical comments could be made from time to time. For instance, you should note that the threshold o 1 m used for filtering wave height in section 3.1 (Figure 2 and l. 264-266) excludes the swells, and leaves only a very homogeneous population of pure wind waves. This will change a lot of things for the assessment of H/T dependence. Similarly, I would comment the fact that you find better results in section 4.2 when you begin by fitting a copula to wave height and wave period, before nesting it with sea level. Indeed, on the one hand you have two parameters (height and period) describing a single physical phenomenon (sea state) and on the other hand a different physical phenomenon (sea level). See for instance the classification of multivariate analyses proposed by Mazas (2017, 2019). I think that it is not by chance that you get the result, in particular because your sea states are pure wind waves.

One final small remark: indicate in Table 1 that Ali-Mikhail-Haq will be noted AMH in what follows.

---

## Author Response (AR2)

*We thank the two reviewers for their comments that contribute all to the clarity of the paper.*

**Reply to reviewer 1**

Removed in abstract : "We select the copulas with the same tail dependence as data. In the event of an opposite tail dependence structure, we resort to the survival copula."

In abstract, the choice of copula is described before the estimation of the copula parameters.

The Defra method is now mentioned in Introduction L34.

The introduction is largely changed taking into account the pertinent remarks of the reviewer.

**Minor Comments**

The term "normal" with copula is removed in the text.

→ "global warming" is changed with "climate change" in L5.
→ L43 "all the pairs" is replaced by the "set of pairs".
→ L56: we add this model is referred to as a "conditional extreme model" in Tiloy et al. (2020).
→ L60 : we add "they show that the fully nested method of creating hierarchical copulas provides the best results for their case study "compared to Chakak and Koehler (1995) and conditional mixture".
→ L76: "Isovalue lines" is replaced by "contours of equal joint exceedance probability".
→ L116: References are added to confirm the use of the assumption : "The independence assumption is not completely valid when two tuples per day are selected but that is an approximation commonly used".
→ L135: "minimum error" is replaced by "minimum mean error".
→ Word "tum" is suppressed.
→ Equation 35: we add "for cumulative distribution functions $U_1$ and $U_2$"
→ Table 1. : Joe copula and Gumbel copula are corrected.
→ L303. "The simplified Defra method refers to univariate survival functions $FH$ and $F\,S$ of wave height and storm surge. The reason is that coastal engineers usually work with exceedance probability rather than with non-exceedance probability" is replaced by "The simplified Defra method refers to univariate survival functions $FH$ and $F\,S$ rather than cumulative distribution functions of wave height and storm surge as coastal engineers usually work with exceedance probability rather than with non-exceedance probability".
→ L308. The sentence is changed : "The bivariate survival functions $\bar{F}_{HS}$ of table 4.15 of Rock Manual (Ciria et al., 2007) are determined with equation (41)".
→ In Table 2 the copulas with the same type of tail dependence as the sample and therefore candidates to model the dataset are highlighted and the following text is added :"In bold in Table 2 are presented the copulas with a lower tail dependence: Clayton, survival Gumbel and AMH when copula parameter is close to 1. We will come back later to this special property of AMH copula. The Gauss copula has a relatively large likelihood. However, it does not have a correct tail dependence and cannot therefore correctly represent the tail dependence."
→ After Table 4, a Figure 9 is added with the text "We show in Figure 9 that there is a coastal area with a maximal dependence from Concarneau to Port-Bloc (in grey in the figure). There are areas that are the most exposed to wind that comes mainly from the West direction along the French Atlantic Coast."

→ One part of section 4.1 is placed in introduction and the section 4.1 is renamed "methodology".
→ L494: "limit" is replaced by "limitations"
→ Three proposed references are added :

Jane, R., Cadavid, L., Obeysekera, J., and Wahl, T.: Multivariate statistical modelling of the drivers of compound flood events in South Florida, Nat. Hazards Earth Syst. Sci. Discuss., https://doi.org/10.5194/nhess-2020-82, in review, 2020.
Tilloy, A., Malamud, B. D., Winter, H., and Joly-Laugel, A.: Evaluating the efficacy of bivariate extreme modelling approaches for multi-hazard scenarios, Nat. Hazards Earth Syst. Sci. Discuss., https://doi.org/10.5194/nhess-2020-28, in review, 2020.
Caillault, C., Guegan, D.: Empirical estimation of tail dependence using copulas. Application to Asian markets. Quantitative Finance, Taylor & Francis (Routledge), 5, 489 – 501, 2005.

**Reply to reviewer 2**

In section 2 and 2.4: the return period T_AND and probability of joint exceedance are defined with a reference to appendix A. Similarly, this probability is defined in section 2.4.

In section 2.1, the description of the constitution of the sample from the time series is completed following the plan proposed by the reviewer. We have removed the mention to the database Candhis that was used to calibrate the database Anemoc in order to avoid a confusion.

In section 4.2, we add : "with the fully nested hierarchical copula method".

In section 3.1, we add : the method that is proposed here for assessing the sample dependence refers to lower tail dependence. Other methods exist such as the chi-plot proposed by Fisher and Switzer (1985, 2001) and used in coastal analyses by Mazas (2017) for instance.

Concerning the use of the wrong tail, we have clarified the notations and changed the text in subsection 3.3. We have kept the notation of survival copula $\bar{C}$ and survival function $\bar{u}$. We had removed the bar in order to simplify the notations but we consider now that we must keep it in order to clarify the reading of the paper.

In section 3.1, we add : "This threshold of one meter that is used for filtering wave height excludes the swells and leaves only a very homogeneous population of pure wind waves. This treatment removes long wave periods and increases the dependence between wave height and wave period."

In section 4.3, we add : "Wave height and wave period are the most correlated variables. This result is not surprising all the more since we deal with pure wind waves after we have removed the swell."

In Table 1, we add: "Ali-Mikhail-Haq will be noted AMH in what follows".

---

## Author Response (AR3)

We thank again the reviewer for the thorough review of our paper that improves a lot the quality of the document.

All the remarks have been taken into account. Is is marked in blue in the manuscript.

Remarks of the reviewer:

Are the sea level time series detrended to ensure the independent and identically distributed (i.i.d.) assumption is met?

More explanation of the complete Defra method would be desirable on line 302 since it is referred to later on in the text i.e. is the Gauss copula used to estimate the dependence between wave height and storm surge?

The explanation on lines 321-325 is not at all clear. I would also be tempted to restate "In the following section, we use survival copula $C$ and survival function $\bar{u}$. Upper tail dependence and lower tail dependence will be inverted." (lines 260-261) here too, to remind the reader that since survival probabilities are being assessed lower tail dependence corresponds to high wave heights and water levels.

Be careful to differentiate between the class of tail dependence and the 'actual' tail dependence. For instance, the Gumbel survival copula has the same class of tail dependence as the Clayton copula (i.e. only lower tail dependence) but the tail dependence is not necessarily the same since it depends on the value of the copula parameters.

The survival Gumbel copula is also referred to as the Gumbel survival copula. Be consistent!

Equations need to be form part of a sentence. Often this correction can be achieved by moving the full stops to after the equation.

Specific comments

Line 12: Grammar. Perhaps change to "joint distribution functions to their".
Line 32: I think they are "reliable return period estimates" rather than "reliable return periods".
Line 34: Grammar. Add "the" before "simplified".
Lines 46-57: The text on all these lines can form a single paragraph.
Line 50: "Copulas generally aggregate only two random variables." I suggest stating more explicitly that this is only the case because most of the studies carried out 'to date' have considered two variables.
Line 51: Grammar. Remove "a" before "specific".
Line 57: As stated by the other reviewer the "conditional mixture" model needs additional explanation.
Line 61: "The pair copula construction (PCC)" change to "PCC" as the acronym has already been defined.
Lines 58-70: The text on all these lines can form a single paragraph.
Line 71: Grammar. Change "to a" to "for a". Also, a bad result is still a result; consider rephrasing the next part of the sentence too.
Line 73: "best results" as compared to which other approaches/models?
Line 74-82: The text on all these lines can form a single paragraph. Also, add an opening sentence summarizing the paragraph e.g. "The paper is divided into three parts.".
Line 81: Consider changing "our practical applications of coastal engineering" to "our coastal engineering based applications" or similar.
Line 86: Replace "he" with "is denoted by" or similar.
Line 163 &166: Consider changing "two-to-two" to "pairwise".
Line 245: Replace "it" with "tail dependence". The next sentence does not make sense; consider removing.
Lines 256 & 260: Replace "it" with "the copula".
Table 1 and Table A1: Typo. Change "Franck" to "Frank".
Line 265: "Gauss" is also an abbreviation requiring a definition.
Line 266: "deal with" is too colloquial.

Line 306: Add "dependence factor " before "FD". As stated by the other reviewer what do the values of 25 and 20 correspond to e.g. Weak dependence? Which corresponds to stronger dependence? Why does Kergadallan (2013) recommend a minimum value of 25?

Line 268-280: The text on all these lines can form a single paragraph.

Line 334: Clarification required here. "as the Gumbel copula has an upper tail dependence, the use of its survival copula is recommended" By recommended do you mean recommended over the Clayton copula? Or recommended as another copula which maybe appropriate given the observed tail dependence?

Line 319: "Since the sample has a tail dependence, it should be known whether it has a lower tail dependence or an upper tail dependence." Not clear!

Line 341: Clarification required here. "We will select the survival copula with the largest likelihood among those which possess the same class of tail dependence as the sample." Is this more accurate?

Line 344: Consider ending the sentence after Student and discussing the lower tail dependence of the AMH in the next sentence. The text currently implies that all of the listed copulas only have lower tail dependence when their parameter(s) are close to 1.

Line 346: Remove "correct" before "tail".

Line 361: "Gumbel survival copula is as large as the log-likelihood of the Clayton copula." Not true, it is smaller! Consider adding "approximately" or similar before large.

Line 364: "is therefore as suitable as the Clayton copula." Again, not sure this is accurate as in the next section it is shown that the Clayton is the most suitable. Perhaps say "potentially as suitable".

Line 386-399: Condense the text on these lines currently there is too much repetition of the line "The points obtained by the XX copula come close to the bisector.".

Line 399: Grammar. "We therefore reestablish a right tail dependence which gives correct results." Rephrase and as stated in the general comments it is the appropriate class of tail dependence.

Figure 8: Shorten caption e.g. Joint exceedance probability obtained with a) Clayton copula (0.38) for Saint Malo b) with Clayton copula (0.74) for Le Havre with tide for return periods of 10, 100 and 1000 years.

Line 438: Remove "a" before "lower".

Line 438: Reorder. "We can therefore conclude that the Clayton copula is the most appropriate copula for our application. For this purpose, the Table 4 gives the parameters of the different sites." Move these two sentences to the end of the paragraph.

Line 446: Grammar. "There are areas" not clear.

Line 453: The use of the term "parameters" is a little confusing as parameter refers to the parameter of the copula elsewhere in the manuscript. Could it be "bivariate copula with a single"? A bivariate copula by definition models the dependence between two variables.

Line 454: "Unlike Corbella (2013) we introduce two parameters". More explanation required.

Line 462: "For all three combinations, the Clayton copula still has the largest maximum likelihood value. In addition, we find that for the combination (H, T) the log-likelihood is significantly higher. The parameters (H, T) are therefore the most correlated parameters." The survival Gumbel copula has a highest maximum likelihood for the (H,T) case. Also, the loglikelihood does not most correlated parameters.

Line 488-500: Shorten and arrange into a single paragraph.

Line 505: "generally good". Too vague. Do you mean the best of the four fitted trivariate copulas?

Line 508: Rephrase e.g. "As expected, the fit of the single parameter Archimedean copula is …".

Table 8: Define "KHI-2" and "KS".

Line 525: "In present practice, ". Please make it clear that joint probabilities are not calculated like this in all cases.

Line 659: "site specific". Given the next sentence it seems they are more "region specific"?

Line 482: "There exist four families". There are more than four families of copulas. Perhaps "Four of commonly applied copulas families are the …" Also, the "Archimedeans, Elliptics" do not need to be plural.

Table A1: Remove an erroneous bracket in the first column of the Joe copula row. Also, please double check the formulae in the Frank copula row.